# Analysis of systemic effects of dioxin on human health through template-and-anchor modeling

Carla M. Kumbale[1¤], Qiang Zhang[2]*, Eberhard O. Voit[1,3]*

1 Wallace H. Coulter Department of Biomedical Engineering, Georgia Institute of Technology and Emory University, Atlanta, Georgia, United States of America, 2 Gangarosa Department of Environmental Health, Rollins School of Public Health, Emory University, Atlanta, Georgia, United States of America, 3 Department of Biological Sciences, University of Texas at Dallas, Richardson, Texas, United States of America

¤ Current Address: Pfizer Inc., Cambridge, Massachusetts, USA
* Eberhard.Voit@UTDallas.edu (EOV), qiang.zhang@emory.edu (QZ)

## Abstract

Dioxins are persistent environmental pollutants known for their multiple health effects, from skin rashes to liver dysfunction, reproductive toxicity and cancer. While the hazards of dioxins have been well documented, the challenge of developing a comprehensive understanding of the overall health impact of dioxins remains. We propose to address this challenge with a new approach methodology (NAM) consisting of a novel adaptation of the Template-and-Anchor (T&A) modeling paradigm. Generically, the template model is defined as a high-level coarse-grained model capturing the main physiological processes of the system. The variables of this template model are anchor models, which represent component sub-systems in greater detail at lower biological levels. For the case of dioxin, we design the template to capture the systemic effects of dioxin on the body's handling of cholesterol. Two new anchor models within this template elucidate the effects of dioxin on cholesterol transport in the bloodstream and on sex hormone steroidogenesis and the menstrual cycle. A third anchor model, representing dioxin-mediated effects on cholesterol biosynthesis via the mevalonate pathway, had been developed previously. The T&A modeling paradigm enables a holistic evaluation of the impact of toxicants, which in the future may be translated into a powerful tool for comprehensive computational health risk assessments, personalized medicine, and the development of virtual clinical trials.

## Author summary

Dioxin is a ubiquitous, persistent pollutant that can have a range of health effects from skin rashes to reproductive problems, liver toxicity and cancer. To characterize and understand pressing health risk problems like those caused by dioxin, the field of chemical toxicity testing has been encouraging the search for "new approach methodologies" (NAMs) that reduce the use of animal experiments. Here, we propose a computational NAM to assess how the diverse effects of dioxin on cholesterol handling in different

**Data availability statement:** Modeling details and computer code can be found in the supplements and on GitHub.com/LBSA-VoitLab/TCDD_Chol, the model is available in Biomodels (MODEL2502110001).

**Funding:** This work was funded in part by NIEHS HERCULES grant P30ES019776-05 and by NIEHS Superfund Research grant P42ES04911. CK and EOV received partial salary from grant P30ES019776-05. QZ received partial salary from grant P42ES04911. The funders had no role in study design, data collection and analysis, decision to publish, or preparation of the manuscript.

**Competing interests:** The authors have declared that no competing interests exist.

organs can be quantified in an organism-wide manner. To this end, we use a novel adaptation of the Template-and-Anchor (T&A) modeling paradigm. The template captures the systemic effects of dioxin on the body's handling of cholesterol, while three anchors explore the impact of dioxin on cholesterol biosynthesis, cholesterol transport and metabolism in the bloodstream, and on female sex hormone synthesis. The T&A modeling strategy permits an overall evaluation of the impact of dioxin, which in the future may be translated into a potent tool for computational health risk assessments, personalized medicine, and the development of virtual clinical trials.

## 1. Introduction

Dioxins are persistent organic pollutants (POPs) that are exceptionally stable and therefore remain in the environment for a long time. In the human body, they often exhibit decade-long half-lives and therefore tend to accumulate in various tissues throughout life. Considering that the World Health Organization (WHO) reported worldwide exposure, dioxins present a potentially severe, global problem [1]. The most toxic among numerous congeners of dioxins is 2,3,7,8-tetrachlorodibenzo-p-dioxin (TCDD), where moderate to severe exposure is known to cause adverse health effects from skin disorders and hormonal disruption to reproductive and developmental dysfunction, liver damage, and cancer [2]. In the following, we will use the more colloquial term "dioxin" as referring primarily to "TCDD."

At the molecular level, dioxin alters the expression of a wide range of genes, including prominently those involved in cholesterol biosynthesis, lipoprotein transport, and steroidogenesis through the estradiol biosynthesis pathway. Many of these alterations are mediated by the aryl hydrocarbon receptor (AhR) [3], which is generally recognized as a sensor for xenobiotic molecules.

Although the hazards of dioxin are well documented in a rich literature, developing a comprehensive understanding of its systemic health impacts remains challenging. Given the numerous contributing factors and their complex interactions, it is appealing to attempt capturing the overall effects of dioxin with computational methods. This pursuit aligns with the new trend toward "New Approach Methodologies" (NAMs) within the toxicological community. These NAMs can be defined as any molecular, analytical or computational method "that when used alone, or in concert with others, enables improved chemical safety assessments through more protective and/or relevant models and, as a result, contributes to the replacement of animals" [4]. Of note here is that computational approaches are included in the envisioned repertoire of desirable NAMs [5].

While it is easy to fathom that computational methods are well suited to capture the many aspects of systemic toxicological assessments, the broad spectrum of effects and the multiscale nature of the impact of dioxin on human health poses a challenge for the development of any comprehensive computational models (*e.g.*, [6–10]). The primary issue is the significant increase in the number of variables, processes, and parameters that must be carried across various temporal and spatial scales. Additionally, defining the exact linkages among these scales poses a substantial hurdle as direct experimental or clinical evidence is often scarce. The complexity of these tasks translates into formidable computational demands and poses difficulties for model validation and standardization. In particular, simulating the system dynamics across various scales tends to be computationally intensive, uncertainties are likely to be compounded, and validating multiscale models often necessitates unique criteria for each level, as well as the overall system. Finally, the lack of standardized methodologies across different domains complicates the analysis, comparison, and interpretation of multiscale systems.

In response to the quest for NAMs, and embracing the complexity posed by the multiscale nature of the problem, we analyze here the organism-wide effect of dioxin on cholesterol handling with a computational approach based on the concepts of Template-and-Anchor (T&A) models [11]. In contrast to the original usage of this paradigm, namely as a tool for identifying biological design principles [11,12], we use the T&A approach simply as a means for organizing details, data, and other pertinent information into a manageable computational structure that is conceptually and practically tractable and flexible enough to facilitate substitutions of sub-models with alternative models in a simple and efficient manner.

The T&A approach uses two types of models that are tightly connected. The template serves as a high-level, coarse-grained model that focuses on the main physiological components of a system and contains only a modest number of variables and their interactions. In contrast, anchor models are more detailed and describe specific biological subsystems or processes in much finer granularity.

As with any mathematical or computational model, the T&A approach is most certainly no panacea; we describe some of the limitations in the discussion. Nonetheless, it allows us to dissect health risk assessments of dioxin into manageable components.

In previous work, we presented one anchor model capturing dioxin-mediated effects on cholesterol biosynthesis via the mevalonate pathway [13]. Here, we introduce the overall template model, two additional anchor models, as well as their relationships to each other. One anchor elucidates the effects of dioxin on cholesterol transport through the plasma, while the other demonstrates the impact of dioxin on ovarian steroidogenesis, which utilizes cholesterol as the source material, and its subsequent effects on the menstrual cycle. Once these anchor models have been analyzed, we demonstrate how the results from these analyses are used to inform the template model. The template model itself is designed to capture the overall impact of dioxin on the body's cholesterol handling and subsequent ramifications for steroidogenesis in the ovary.

## 1.1. Biomedical background

### 1.1.1. Global effects of dioxin on cholesterol handling.
As documented in the *Comparative Toxicogenomics Database* (CTD) [14], dioxins, and especially TCDD, affect the expression of a multitude of genes. In many cases, the direct sequelae of these alterations are unknown, but there is substantial information regarding three target areas associated with cholesterol handling: cholesterol biosynthesis through the mevalonate pathway; lipoproteins and their role of cholesterol transporters; and the reproductive system, especially the effects of dioxin on ovarian hormones. These three target areas are of course connected, leading to a complex web of effects of dioxin, which is sketched below. Table 1 displays the direct qualitative effect of dioxin on the expression of genes associated with the organ systems in the anchor models.

### 1.1.2. Effects of dioxin on cholesterol biosynthesis.
Hepatic cholesterol biosynthesis was described in detail in [13]. At its core is the mevalonate pathway, which consists of a highly regulated multistep process (**Fig 1**). It encompasses a metabolic cascade that begins with the synthesis of HMG-CoA (3-hydroxy-3-methyl-glutaryl-coenzyme A) and progresses through several enzymatic steps, ultimately leading to the production of cholesterol. The mevalonate pathway is tightly controlled through negative feedback regulation exerted by the transcription factor SREBP (Sterol Regulatory Element-Binding Protein), which ensures cholesterol homeostasis. Additionally, the anchor model describing this pathway includes the conversion of cholesterol into bile acids, which constitute a significant route of cholesterol utilization, the storage of cholesterol in the liver as cholesterol esters, and the exchange of cholesterol between the liver and plasma. Dietary cholesterol intake also influences this

**Table 1. Genes, enzymes, and proteins positively (↑) or negatively (↓) affected by dioxin in the anchor models, according to the Comparative Toxicogenomics Database [14].**

| Gene | Enzymes/Proteins | Impact of Dioxin on Expression | Organisms | Sources |
|---|---|---|---|---|
| HMGCS1 | HMG-CoA synthase | ↓ | Homo sapiens Mus musculus | Iqbal K (2021) |
| HMGCR | HMG-CoA Reductase | ↓ | Homo sapiens Mus musculus | Boutros PC (2009) |
| FDPS | Farnesyl diphosphate synthase | ↓ | Homo sapiens Mus musculus | Fracchiolla NS (2016) |
| SQS | Squalene-synthase | ↓ | Homo sapiens Mus musculus | Hurst CH (1998) |
| LSS | Lanosterol-synthase | ↓ | Homo sapiens Mus musculus | Labaronne (2017) |
| APOA1 | Apolipoprotein A1 | ↑ | Mus musculus | Angrish (2013) |
| LCAT | Lecithin-cholesterol acyltransferase | ↑ | Mus musculus | Angrish (2013) |
| LDLR | Low-density lipoprotein receptor | ↑ | Mus musculus | Nault R (2017) Angrish (2013) |
| LPL | cytochrome p450 family 7 subfamily A member 1 | ↑ | Homo sapiens Mus musculus | Fader KA (2015) Nault R (2017) |
| CYP17A1 | cytochrome p450 family 17 subfamily A member 1 | ↓ | Mus musculus | Fader KA (2015) |
| HSD3B1 | hydroxy-delta-5-steroid dehydrogenase, 3 beta- and steroid delta-isomerase 1 | ↓ | Homo sapiens Mus musculus | Saurat JH (2012) |
| CYP19A1 | cytochrome p450 family 19 subfamily A member 1 | ↓ | Homo sapiens Mus musculus | Baldrige MG (2015) |
| HSD17B1 | hydroxysteroid 17- beta dehydrogenase 1 | ↓ | Mus musculus | Karman (2012) |
| CYP1B1 | cytochrome p450 family 1 subfamily B member 1 | ↑ | Homo sapiens Mus musculus | Iqbal K (2021) |
| COMT | catechol-O-methyltransferase | ↑ | Mus musculus | Boutros PC (2009) |
| CYP11A1 | cytochrome p450 family 11 subfamily A member 1 | ↑ | Homo sapiens Mus musculus | Baldrige MG (2015) |

pathway, reflecting the impact of external cholesterol sources on internal biosynthesis and overall cholesterol balance.

**1.1.3. Effects of dioxin on lipoprotein dynamics.** Lipoproteins serve as carriers of lipids in the bloodstream. They consist of a core of cholesterol esters and triglycerides surrounded by a shell of phospholipids, free cholesterol, and apolipoproteins [16]. Lipoproteins are typically classified into one of five groups: chylomicrons, very low-density lipoproteins (VLDL), VLDL remnants (IDL), low-density lipoproteins (LDL), and high-density lipoproteins (HDL) [16]. The literature on these lipoproteins is enormous.

A simplified depiction of lipoprotein metabolism and transport is shown in **Fig 2**. This diagram serves as the direct basis for constructing the anchor model (see *S1 Text, Section 2.3*).

In the liver, cholesterol is synthesized *de novo* or taken up from the bloodstream via low-density lipoprotein receptors (LDL-R). Hepatic cholesterol can then be converted to cholesterol esters and stored, a process catalyzed by the enzyme acetyl-CoA acyltransferase (ACAT), or it can be incorporated into VLDL for secretion into the bloodstream. Once in the bloodstream, VLDL is hydrolyzed by lipoprotein lipase (LPL) leading to the formation of IDL. IDLs are either taken up by the liver or further catabolized to LDLs by an enzyme called hormone sensitive lipase (HSL). LDLs are then transported into the liver or peripheral cells where they are utilized for various cellular processes, including membrane synthesis and steroid

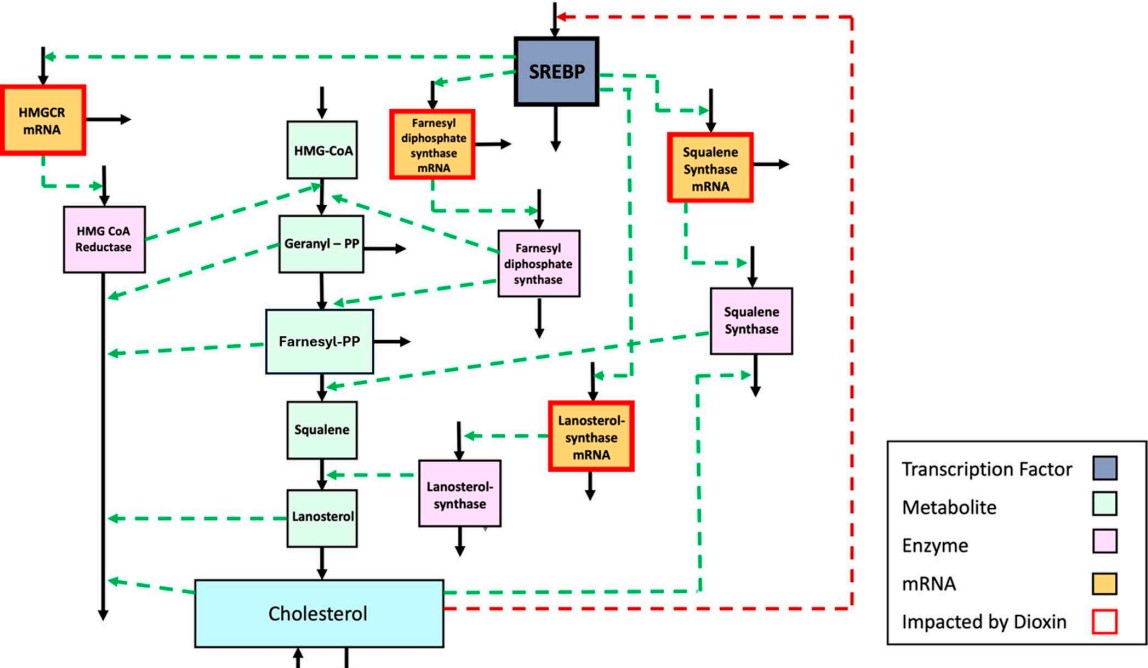

**Fig 1. Hepatic cholesterol biosynthesis through the mevalonate pathway as one of the anchor models.** The various components of the biosynthetic mevalonate pathway system are color-coded by type, as indicated. Metabolic reactions, transport steps, and the production of mRNAs and proteins are shown as solid lines, while regulatory signals, representing feedback inhibition (red) and modulation of enzyme activity [15], are given as dotted lines; mRNAs whose transcription is affected by dioxin, are indicated by boxes outlined in red. The key transcription factor SREBP binds to sterol regulatory elements (SREs) in the promoter regions of its target genes. It controls the activity of mevalonate pathway via these target genes, among which HMG CoA (3-hydroxy-3-methyl-glutaryl-coenzyme A) reductase is considered the rate-limiting step of the pathway, even though control of the pathway is distributed. Adapted from [13].

hormone production [17]. LDL-R facilitates the uptake of VLDLs, IDLs, and LDLs into the liver [18]. Additionally, HDL acquires free cholesterol from the cell membranes of peripheral tissues, which allows it to be esterified by lecithin cholesterol acyltransferase (LCAT). It is believed that this esterification of cholesterol by LCAT promotes the net removal of cholesterol from peripheral tissue and incorporation into HDL [19]. HDLs then transport the excess cholesterol from peripheral tissues back to the liver for excretion, storage or recycling in a process called reverse cholesterol transport [20]. This process is facilitated by scavenger receptor class B type 1 (SR-B1), which mediates selective uptake of cholesterol from HDL particles by hepatocytes [21].

According to CTD [14], dioxin strongly influences the lipoprotein transport pathway. Specifically, by interacting with AhR, dioxin can alter the expression of genes that are integral to the production, conversion, and uptake of lipoproteins. Genes that are pertinent for our model and impacted by dioxin in rodents and/or humans are presented in Table 1.

**1.1.4. A brief overview of the female reproductive system.** The female sex hormones 17β-estradiol (E2) and progesterone (P4) are synthesized in the ovaries. The steroidogenesis process occurs in the antral follicles during the follicular phase and additionally in the corpus luteum during the luteal phase. Antral follicles are composed of granulosa cells in the inner layers and theca cells in the outer layers. These two types of cells work together to produce sex steroid hormones according to the so-called two-cell-two-gonadotropin principle [22], according to which the theca cells utilize cholesterol to make progesterone and

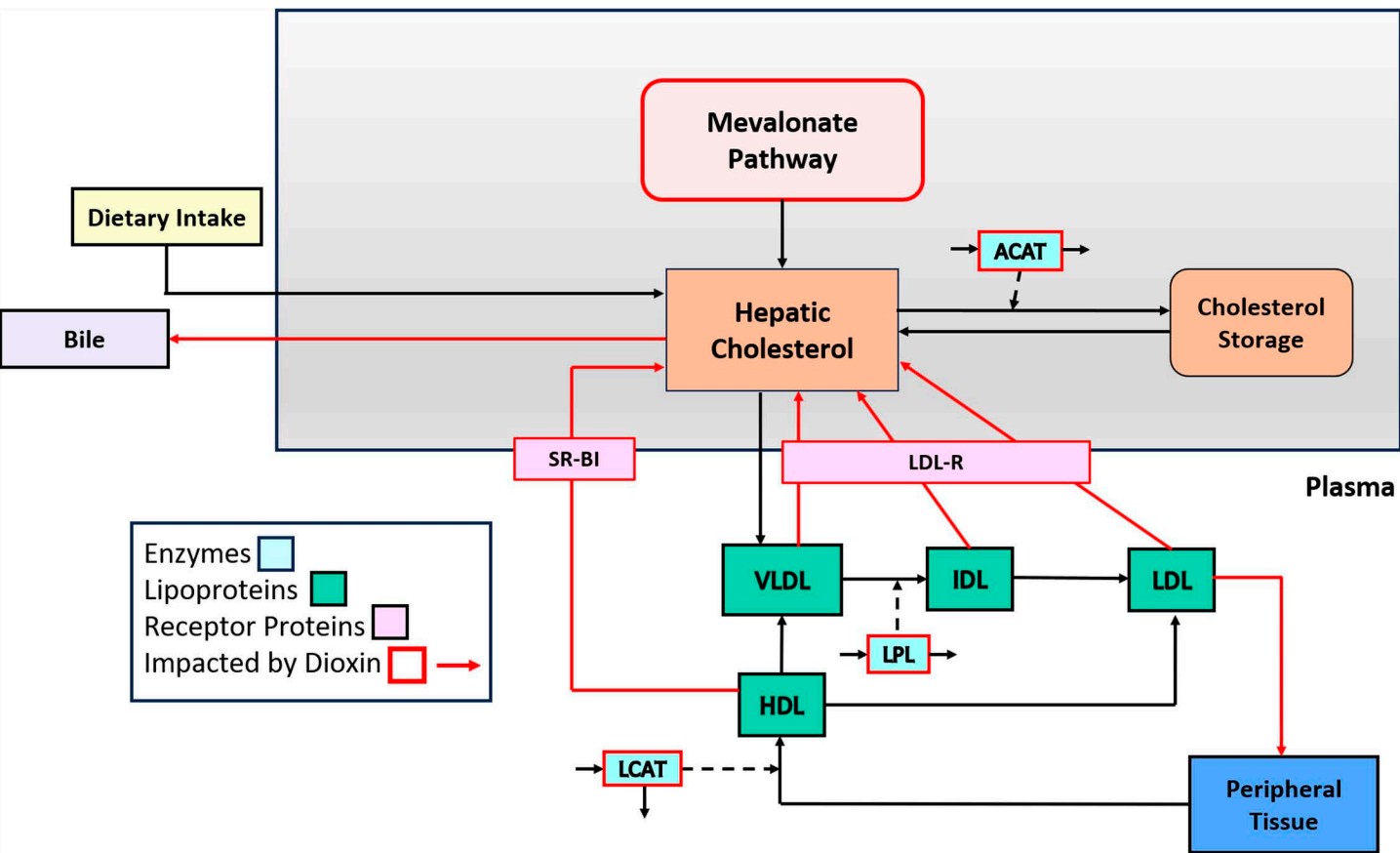

**Fig 2. Overview of cholesterol metabolism and transport pathways.** The diagram illustrates the processes of organism-wide cholesterol handling, and especially the involvement of the lipoproteins VLDL, IDL, LDL, and HDL, which are responsible for the transport of cholesterol between the liver and the peripheral tissues. The receptor proteins SR-BI and LDL-R, along with enzymes such as LPL and ACAT, are also depicted. The red boxes and arrows indicate the impact of dioxin on specific processes.

androgen, and androgen diffuses into the granulosa cells where it is converted to estrogen. The steroidogenesis in the corpus luteum works in a similar fashion, involving luteinized, differentiated theca and granulosa cells [23].

The initial rate-limiting step in the steroidogenesis pathway (see **Fig 3** for a simplified representation) is the transport of cholesterol from the mitochondrial outer membrane to the inner membrane, which is mediated by the steroidogenic acute regulatory protein (StAR). Subsequently, cholesterol is converted to pregnenolone (P5) by the cholesterol side-chain cleavage enzyme 20,22-desmolase, which is also called CYP11A1 and is located in the inner mitochondrial membrane [24–26]. P5 then moves out of the mitochondrion and into the endoplasmic reticulum, where it is converted to P4 by 3β-hydroxysteroid dehydrogenase (HSD3B1), or to dehydroepiandrosterone (DHEA) by CYP17A1. P4 can either be released into the bloodstream or partially transformed into androstenedione (A4) by the enzyme CYP17A1. Additionally, DHEA can be converted into A4 by the enzyme HSD3B1. A4 is then transformed into testosterone (T) by HSD17B1 and T is converted to E2 by the aromatase CYP19A1. An alternative path to the last two steps is that A4 is first converted to estrone (E1) by CYP19A1, and E1 is then converted to E2 by HSD17B1. E2 is eventually secreted into the blood circulation.

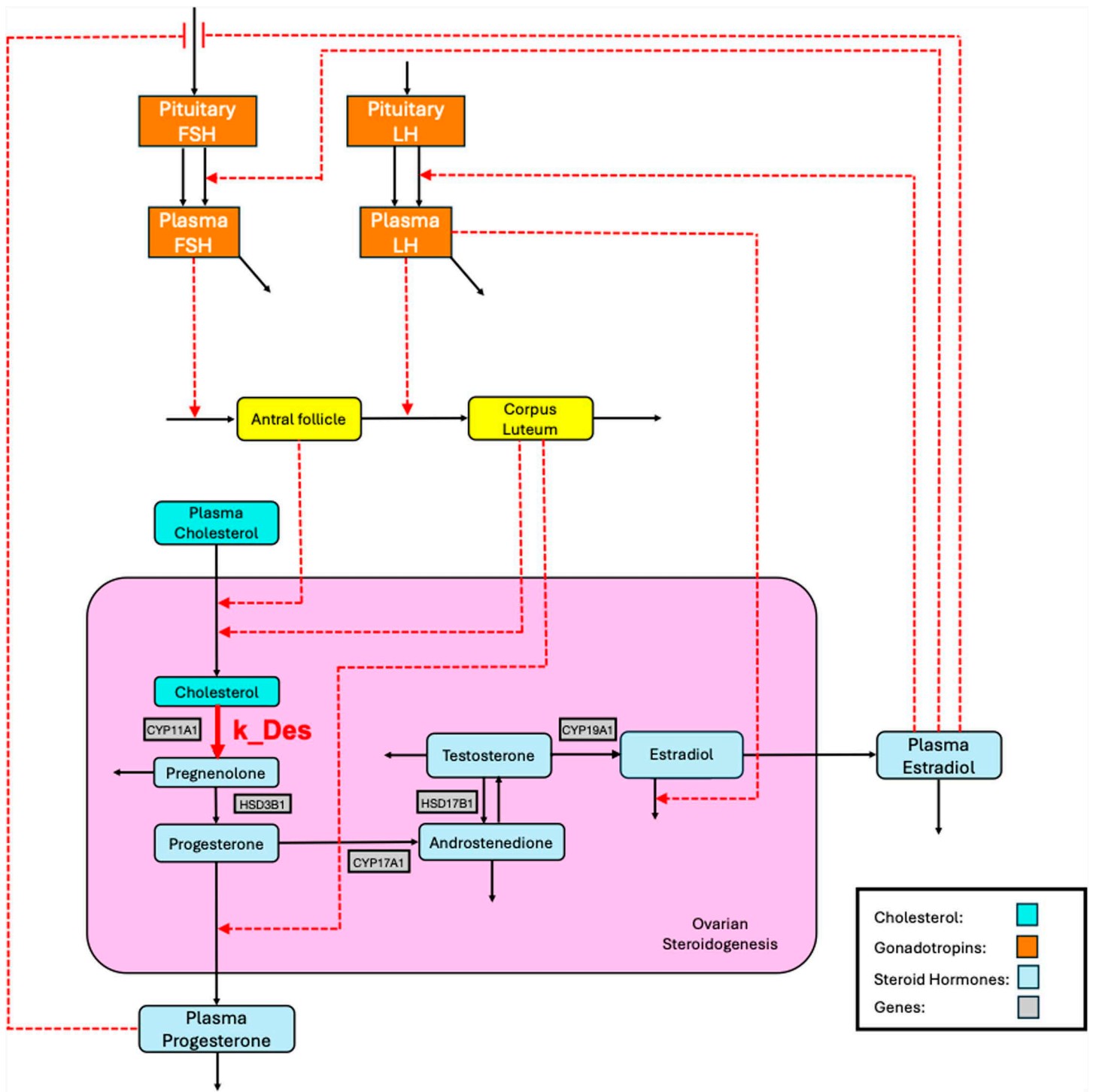

**Fig 3. Anchor model of female steroidogenesis, describing the estradiol synthesis pathway, its regulation, and perturbation by dioxin.** Molecules are color-coded by type, as indicated. Solid lines show reactions and transport steps, with the red solid line signifying the 20,22-desmolase reaction (k_Des), which is affected by dioxin. Dashed lines represent feedback regulation (red) from estradiol and progesterone on the synthesis and release of FSH and LH hormones from the pituitary to plasma, exerting control over the development of the antral follicles and the corpus luteum. Key genes are displayed in grey rectangular boxes.

The overall structure of the model is illustrated in Fig 3. It contains the essential steroid metabolites and feedback interactions between the pituitary gonadotropins FSH and LH and the ovarian hormones E2 and P4, the antral follicle and corpus luteum dynamics, and the steroidogenesis pathway. The steroidogenesis pathway contains several enzymatic steps. The initial rate-limiting step is marked with k_Des (desmolase; see Fig 3). All steroid conversion steps were modeled as mass-action processes, possibly with regulation in power-law format, without explicit concern for the specific enzymes involved.

FSH stimulates the growth of the antral follicle. As the follicle increases in mass, its steroidogenic activity increases because more theca and granulosa cells are formed. This increase in steroid synthesis during follicle growth is implemented in the model by increasing the supply of plasma cholesterol as the substrate to the first step of the steroidogenesis pathway. As E2 in the plasma increases, it inhibits the production of pituitary FSH, causing FSH to decrease. Once the increase in plasma E2 reaches a certain threshold level, it triggers the release of LH, which was so far stored in the pituitary, into the blood circulation, producing a plasma LH surge. A similar surge is triggered in FSH.

The E2 threshold was implemented here by an intermediate signaling step S that functions as a bistable switch in response to E2 (see *S1 Text, Section 2.3*). The LH surge converts the antral follicle to the corpus luteum, which eventually regresses. The corpus luteum drives the steroid synthesis also by increasing the supply of plasma cholesterol. In addition, the corpus luteum promotes the release of P4 to the circulation, causing a P4 rise in the luteal phase.

**1.1.5. Effects of dioxin on the female reproductive system.** Sufficient exposure to dioxin can cause a variety of adverse reproductive outcomes, including altered sex hormone dynamics, anovulation, reduced fecundity, delayed fetal growth and development, and endometriosis [27]. Dioxin likely has multiple toxicity targets within the female reproductive system of a variety of species, but the most common appears to be inhibition of female steroid hormone production and disruption of the reproductive cycle. For instance, rhesus monkeys that had been exposed to 500 ppt dioxin in the diet for six months exhibited decreases in both E2 and P4 levels [28]. Similarly, chronic exposure to dioxin induced a dose-dependent decrease in the E2 level of proestrus in rats [29]. Dioxin also caused E2 to decrease in human luteinizing granulosa cells [30]. In mouse antral follicles, studied *in vitro*, dioxin exposure suppressed the secretion of P4, A5, T, and E2 over a period of 96 hours, showing a concentration-dependent pattern of inhibition [31,32].

Exposure to dioxin in rats can interfere with the estrous cycle, resulting in an extended duration of the diestrus phase [33]. Similarly, rhesus monkeys exposed to dioxin through their diet exhibited a trend toward slightly longer menstrual cycle lengths, increasing from 27.3 ± 4.3 to 27.8 ± 3.9 days, although this average increase was not statistically significant [28]. Epidemiological studies showed that the menstrual cycle length of women living in the dioxin-polluted area of Seveso, Italy was positively associated with serum dioxin concentrations and that women with high serum dioxin concentrations experienced longer times to pregnancy and increased risk of infertility [34]. Serum levels of estrogenic polychlorinated biphenyls, some of which are dioxin-like compounds, were significantly associated with increased menstrual cycle length, which was prolonged by up to 3 days [35].

While animal and *in vitro* studies both show that exposure to dioxin generally leads to suppression of ovarian E2 secretion, the steroidogenic enzymes altered by dioxin appear to vary depending on species and experimental conditions. In human luteinizing granulosa cells, dioxin causes E2 to decrease without changing either the aromatase protein or its enzyme activity [36]; however, there is a decrease in CYP17A1 protein abundance and activity [37]. In these cells, dioxin reduces mRNA expression of CYP11A1 and CYP19A1.

Similarly, in granulosa cells from young or adult rats, dioxin lowers mRNA expression of the key enzyme CYP11A1 (20,22-desmolase) and of CYP19A1 [38]. In rats chronically exposed to dioxin, CYP17A1 mRNA expression is significantly inhibited while the expression of genes of other steroidogenic enzymes is not altered [29]. In mouse antral follicles, dioxin inhibits 20,22-desmolase, CYP17A1, HSD17B1, and CYP19A1 mRNA expression, and increases HSD3B1 mRNA expression; however, at the protein level only the abundance of HSD17B1 is significantly reduced [32].

In summary, sufficient exposure to dioxin can dysregulate several steroidogenic enzymes, inhibit steroid hormone production, and extend or disrupt the ovarian cycle.

## 2. Methods

A generic challenge of any modeling effort is that the true functional forms capturing the dynamics of complex biomedical systems are unknown and hardly identifiable with experimental means [39]. As a potent solution to this conundrum, we use canonical modeling with methods of Biochemical Systems Theory (BST) [39–45] and its most prominent special case, Mass-Action Kinetics (MAK). The resulting ordinary differential equations (ODEs) contain two types of parameters: (1) non-negative rate constants ($v_{ij}$) of production and degradation, and (2) kinetic orders ($a_j$), whose signs directly align with the modeled function. In MAK, the kinetic orders are non-negative integers (usually, 0, 1, or 2), while they may be any real numbers in general BST systems. Positive kinetic orders in BST identify augmenting or increasing influences, whereas negative kinetic orders signal inhibitory effects. BST is mathematically rigorous, as it is directly based on Taylor's approximation theory of numerical analysis.

### 2.1.  Template model

The T&A paradigm provides a structured framework for exploring both the overall dynamics of a complex system and the intricate details of its underlying sub-systems. The high-level dynamics of a system is captured in the template model. For the problem space of dioxin and its health effects, one could imagine a variety of template models. Due to the critical importance of cholesterol, we structure our template around the handling of cholesterol by different organ systems in the body. A pertinent organism-wide diagram is shown in **Fig 4**.

The dependent (state) variables of a template model, rather than being conventional model variables like molecular concentrations, are the anchors, which themselves are models at a lower level of organization and collectively represent processes, such as cholesterol biosynthesis. In our template model, the interaction arrows indicate the handing over of cholesterol (in some form) or direct or indirect signaling events modulating cholesterol-associated processes within the various boxes. We will discuss these conceptual roles of variables and arrows later in more detail.

There is no general prescription for which variables should ultimately by expanded into full anchor models. We considered as anchors three systems that correspond to the most prominent processes of cholesterol handling:

1. Cholesterol biosynthesis in hepatocytes, combined with cholesterol storage through the reversible formation of cholesterol esters, and transport among liver, plasma, and peripheral tissues. This anchor model was discussed elsewhere in detail [13].

2. The pathway system governing the roles of the lipoproteins LDL, HDL, VLDL and IDL, which transport cholesterol within the bloodstream.

3. The dynamics of endocrine processes pertaining to the synthesis of progesterone, testosterone, estradiol, and related steroids in the gonads.

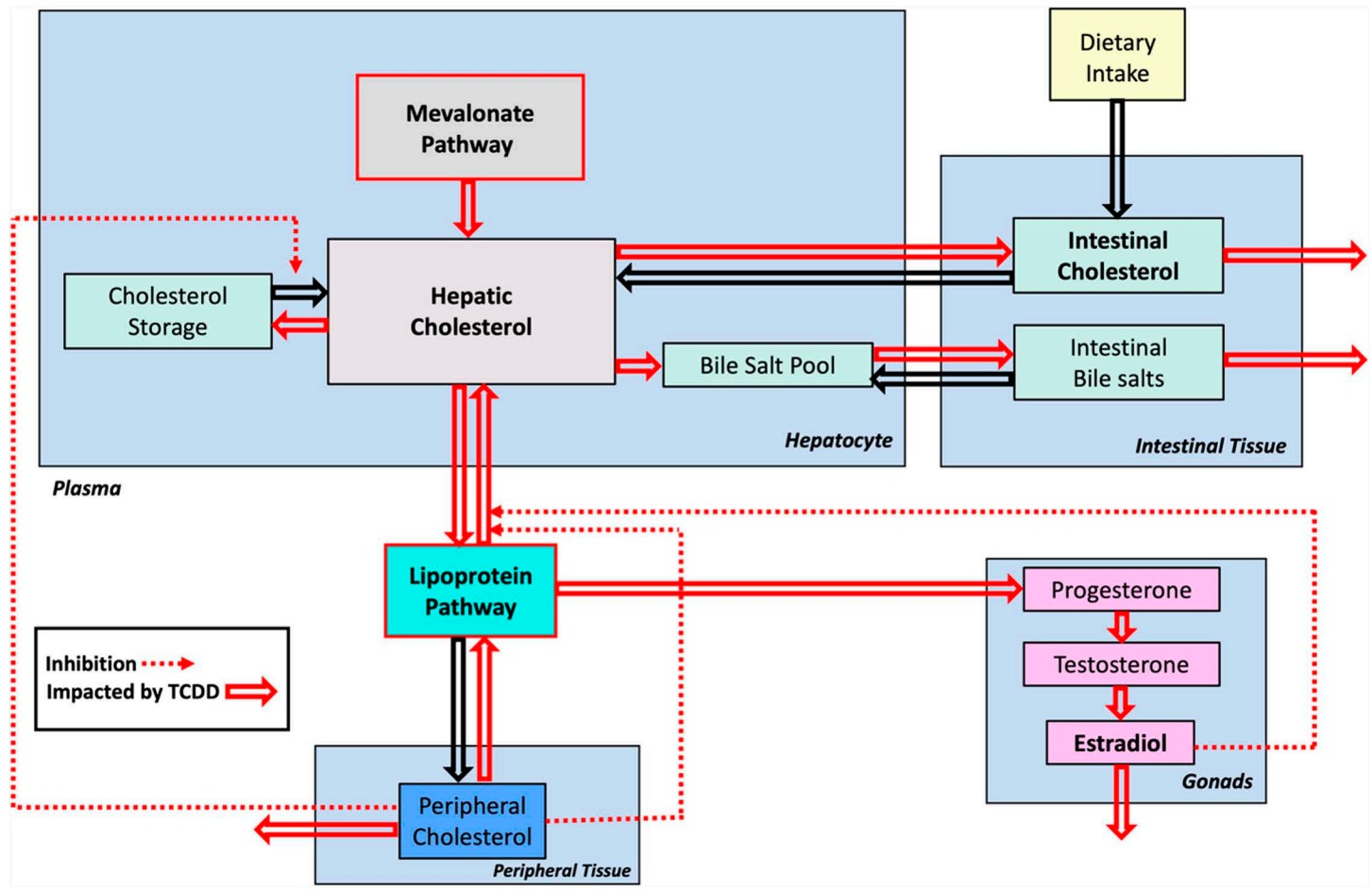

**Fig 4. Template model of dioxin exposure affecting organism-wide cholesterol handling at a systemic level.** Processes affected by dioxin exposure are indicated by red arrows. Adapted from [13].

If deemed of benefit, one could explicitly consider further anchor models, which are presently not addressed or subsumed in the three anchors above. Examples are cholesterol storage and utilization in peripheral tissues, which at this point are kept as template variables that are not expanded into full anchor models. This decision illustrates the flexibility of the T&A approach to zoom in or out from a different viewpoint.

The analysis of the template model allows a global assessment of the effects of dioxin on the human body, while analysis of the anchors captures the molecular details and mechanisms associated with cholesterol in the various organ systems. The results from analyzing the anchors and the effects of dioxin will ultimately be used to inform the parameterization and modulation of the template model, as described later.

**2.1.1. Mathematical formulation of the template model.** The first step of about any model design and analysis is the creation of a block diagram consisting of boxes and interaction arrows [46,47], and T&A models are no exception. The T&A diagrams resemble diagrams that are common in control theory and biomathematics for modeling natural and engineered dynamical systems. The block diagrams in control theory schematically depict how an input signal to a system undergoes various transformations through blocks of transfer functions, including the potential forward flow of a signal or

feedback signals. The transfer functions are typically modeled in the signal domain as the Laplace transforms of the mathematical functions that transmit an input signal to an output signal. Due to the assumed linearity of the engineered system, the transformations of an input signal to an output in a block diagram are relatively simple and may include feedback signals or the forward flow of information. Such signals could also apply to biomedical systems, but these are almost always nonlinear, thereby excluding Laplace transforms from the repertoire of applicable methods. Within the T&A context, both template and anchor models capture the dynamic behavior of processes that mutually interact in complicated ways.

Of importance in our context is that the variables of the template model, when zoomed in, are fully functional, more finely grained dynamic models, which serve as anchors. These anchor models provide a much more detailed view than typical block diagrams, and a visualization would require something like a nested block diagram, with nonlinear influences flowing up and down the hierarchy of the diagram.

To streamline the implementation of the dioxin model, we slightly simplify and rename the pathway system in Fig 4 toward the diagram in **Fig 5**, which is then mathematically represented with nonlinear ODEs in the format of mass-action or generalized mass action (GMA) systems in BST [40,45,47,48]. Using this format, and given a proper system diagram, developing a corresponding mathematical model is comparatively straightforward. Of particular note is that this strategy is effective even in the face of uncertainties. For instance, an observation like "if the value of Component A increases by this percentage, Component B decreases by this percentage" directly suggests an inverse relationship that can be captured well with a power-law function that constitutes the core of BST.

Recalling that the variables in the template model represent processes, their steady-state values correspond to the steady-state fluxes of the anchor models, which may be directly affected by other template variables. The edges and their mathematical representations capture how the change in one template box (anchor) affects a neighboring box. If a box is the source of a signal, this signal is represented slightly differently than in typical models, because a signal here exerts its effect on other boxes, rather than on arrows, as it is typical in conventional models (**Fig 6**).

We begin the specific set-up of the template model by supposing the absence of dioxin exposure. The model described in the format of GMA equations is presented below. Even though the variables have a different meaning than one is used to, they contain the same two types of parameters: (1) non-negative rate constants ($v_i$) of production and degradation, and (2) kinetic orders ($a_j$), whose signs directly align with the modeled function: activating or augmenting processes are represented with positive kinetic orders, while inhibitory or diminishing processes are modeled with negative kinetic orders.

The GMA representation of the template model in Fig 5 thus has the following format:

**Biosynthesis**

$$
\begin{aligned}
\dot{X}_1 = {} & v_{01} \cdot X_0^{a1} \cdot X_2^{a2} \cdot X_3^{a3} \cdot DR_1 + v_{41} \cdot X_3^{a4} \cdot X_4 \cdot DR_2 \\
& + v_{21} \cdot X_2 \cdot X_3^{a5} \cdot DR_3 \\
& - v_{14} \cdot X_0^{a1} \cdot X_1 \cdot DR_4 - v_{12} \cdot X_1 \cdot DR_5 - v_{16} \cdot X_1 \cdot DR_6
\end{aligned}
\tag{1}
$$

**Storage**

$$
\dot{X}_2 = v_{12} \cdot X_1 \cdot DR_5 + v_{20} \cdot X_0^{a6} - v_{21} \cdot X_2 \cdot X_3^{a5} \cdot DR_3
\tag{2}
$$

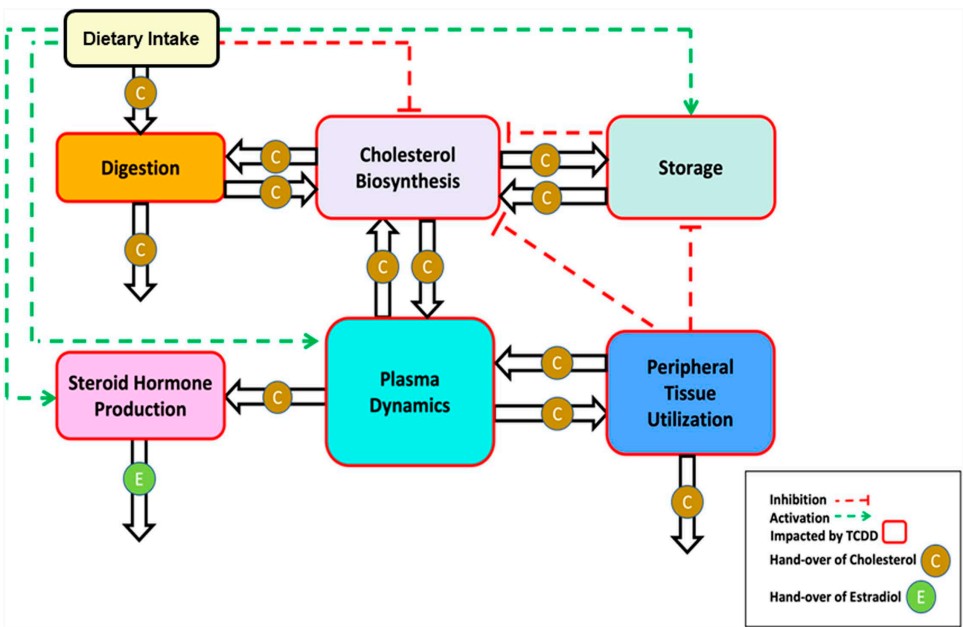

**Fig 5. Simplified template model.** The variables correspond to the processes in Fig 4. Specifically, Dietary intake: $X_0$; Cholesterol biosynthesis: $X_1$; Cholesterol storage: $X_2$; Peripheral cholesterol utilization: $X_3$; Plasma transport and conversion of cholesterol: $X_4$; Steroid hormone production: $X_5$; Excretion: $X_6$. The hollow arrows represent the hand-over of cholesterol from one process to another. Each activation arrow (green, dashed) represents a positive effect of the source process on the target process. For example, dietary cholesterol intake leads to increased cholesterol storage. An inhibition arrow (red, dashed, blunted) indicates that the source process affects the target process. For example, dietary intake of cholesterol reduces the rate of cholesterol biosynthesis. Once it is clear that the variables represent processes, the mathematical implementation in BST follows the same guidelines as for a traditional model: The change in the rate of a process is driven by increasing and decreasing influences, and each term describing these influences is modeled as a power-law term containing a rate constant and those variables (processes) that directly affect the term in question, raised to an appropriate power.

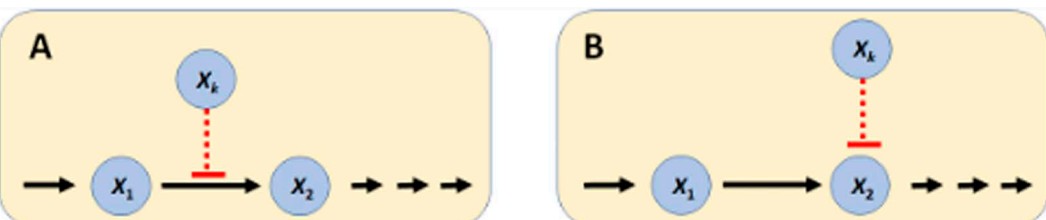

**Fig 6. Signals in conventional models and T&A models.** A: In a conventional model, variables modulate fluxes, either through activation or inhibition. B: In a T&A model, variables represent process systems that can directly affect each other.

### Cholesterol Utilization in Peripheral Tissue

$$\dot{X}_3 = v_{43} \cdot X_4 \cdot DR_7 - v_{34} \cdot X_3 \cdot DR_8 - v_{30} \cdot X_3 \tag{3}$$

### Plasma Dynamics

$$\dot{X}_4 = v_{34} \cdot X_3 \cdot DR_8 - v_{43} \cdot X_4 \cdot DR_7 - v_{41} \cdot X_3^{a4} \cdot X_4 \cdot DR_2$$
$$+ v_{14} \cdot X_0^{a1} \cdot X_1 \cdot DR_4 - v_{45} \cdot X_0^{a1} \cdot X_4 \cdot DR_9 \tag{4}$$

**Steroid Hormone Production**

$$\dot{X}_5 = v_{45} \cdot X_0{}^{a1} \cdot X_4 \cdot DR_9 - v_{40} \cdot X_5 \tag{5}$$

In these equations, each quantity $DR_i$ represents the effect of dioxin on a given process. At the baseline (dioxin absent ↔ TCDD = 0), all $DR_i$ have a value of 1. For different dioxin exposure scenarios, the $DR_i$ are numerically adjusted according to dose-response curves developed in the anchor models. Specifically, the amount of cholesterol handled under exposure to a dioxin dose of interest is extracted from the dose-response relationships obtained from the appropriate anchor models that are to be analyzed separately (see [13] and below). This modified amount of cholesterol is now handed off to other variables of the template model. In other words, dose-response relationships are used as dioxin-dependent modulators of the baseline fluxes under no exposure. They are applied to pertinent template variables, such as the biosynthesis pathway or cholesterol handling in the bloodstream (see Figs 1 and 2).

Once defined as an independent (external) variable, the dioxin dose can be increased incrementally across all known affected processes, where different degrees of dioxin effect can be represented with corresponding fold changes for different fluxes ($DR_i > 1$ or $DR_i < 1$) and magnitudes reflecting augmenting or diminishing effects. Thus, the template model at baseline (TCDD = 0) represents the normal, physiological state and dynamics of the cholesterol system. Once the anchors have been generated, importing dose-response information into the template allows us to gain a more comprehensive understanding of the systemic impact of dioxin on affected processes at the organismic level.

**2.1.2. Parameter determination and model evaluation.** If sufficient information were available regarding cholesterol sharing among organs, the template model could be parameterized from this information. However, such explicit information is lacking. Instead, the input/output relationships, obtained from the separately established anchor models, are used to inform steady-state values of the template variables and the connections between them. Kinetic-order parameters which, for instance, describe the influence of estradiol signaling and peripheral tissue utilization on cholesterol uptake in hepatocytes, as well as hepatic cholesterol storage, are then adjusted to render output consistent with published steady-state concentration data.

Upon calibrating the model parameters to match the steady-state concentrations that have been reported for the anchor models, the output of the template model at baseline (TCDD = 0) exactly reflects the steady-state fluxes in the anchor models. Specifically, the fluxes for biosynthesis, storage, tissue utilization, plasma transport, and steroidogenesis are 18,150, 0.56, 266, 3,266, and 1.34 µg/kg, respectively.

## 2.2. Anchor model of cholesterol biosynthesis

The biosynthesis anchor model was analyzed and discussed elsewhere in detail [13]. It contains as variables the metabolites of the pathway, as well as pertinent mRNAs, enzymes, and the important transcription factor SREBP, which is controlled by the amount of hepatic cholesterol through feedback inhibition (Fig 1). Using methods of BST and MAK, it was relatively straightforward to convert the diagram of the mevalonate pathway into a dynamic mathematical anchor model (see, *e.g.*, [39,41,45]). Literature searches directly or indirectly yielded parameter values, as well as information quantifying the effect of dioxin. Model equations are revisited in *S1 Text*. The biosynthesis model is also available on GitHub (GitHub.com/LBSA-VoitLab/TCDD_Chol) and has been submitted to Biomodels.

As a side note, our analysis [13] actually targeted two models of biosynthesis: One without interactions with plasma, and one where the pathway was combined with a simplified

representation of cholesterol transport between the liver and plasma. This use of alternative anchors illustrates the flexibility of the T&A approach in terms of substitutions of anchors with models that are deemed more appropriate for a specific purpose.

## 2.3. Anchor model of lipoprotein metabolism and cholesterol transport

As with the biosynthesis pathway, we used methods of BST to model the roles of lipoproteins. BST permits a direct conversion of the pathway diagram into symbolic equations [41]. Typical examples are the ODEs for LDL and HDL:

**LDL Cholesterol:**

$$\dot{LDL} = LDL = c_{11} \cdot HDL - c_8 \cdot LDL \cdot DR_7 + c_{21} \cdot IDL \cdot DR_{10} - c_{22} \cdot LDL \tag{6}$$

**HDL Cholesterol:**

$$\dot{HDL} = c_{13} \cdot LCAT^{k_3} \cdot DR_{11} - c_9 \cdot HDL \cdot DR_8 - c_{11} \cdot HDL - c_{12} \cdot HDL \tag{7}$$

Here, all kinetic orders are set to 1, which is typical for mass-action and transport models. The indexed quantities $DR_i$ are placeholders that have a value of 1 at the baseline (TCDD = 0) but allow us later to modify rates according to process-specific dose-response relationships of dioxin exposure.

The complete set of ODEs can be found in *S1 Text*, *Section 2.1*. Computer code in PLAS and MATLAB format is available on GitHub.com/LBSA-VoitLab/TCDD_Chol.

**2.3.1. Parameter determination and model evaluation.** Although the model is fully dynamic, it is evaluated here at the steady state because we are only interested in the long-term effects of dioxin exposure, which implies that the system has had enough time to reach a steady state. As a consequence, one rate in each equation, determining the time scale of the dynamics of the variable, is more or less arbitrary and may be set to some reasonable positive value (see below). The reason is that multiplying a first-order ordinary differential equation with a positive factor, such as 0.1 or 10, causes the dynamics of the equation to run at a tenth of the speed or ten times as fast as the original, respectively. This multiplication does not affect the steady state. Thus, by choosing a non-zero positive multiplier, the speed of the dynamics can be manipulated. Turning this argument around, we cannot identify the value of this multiplier because the only available experimental information for the calibration of the model pertains to the steady state.

The remaining parameters were set such that the model precisely matched the steady-state concentrations in human hepatocytes or plasma, as reported in the literature, namely: hepatic cholesterol: 11,636 μM [49]; VLDL: 776 μM [50]; LDL: 2,586 μM [50]; HDL: 1293 μM [50]; IDL: 388 μM [50]. Upon calibrating the model parameters, the steady-state lipoprotein concentrations obtained from the literature and those computed with the parameterized model were quite similar to each other. Basal protein activity levels were arbitrarily defined as 100, reflecting "normal conditions." Values of other model parameters were based on previously obtained parameters from the mevalonate anchor model [13] and from literature information; they were reasonably adjusted to obtain model outputs consistent with published steady-state concentration data. The remaining parameter values were manually adjusted to align approximately with the expected half-lives of the lipoproteins and enzymes. Finally, we calibrated the percent change in lipoprotein cholesterol concentration under dioxin exposure based on animal studies for a dioxin dose of 30 μg/kg [51]. For simplicity, and due to the lack of specific

information, we then used interpolation to obtain lipoprotein cholesterol concentrations at other doses of dioxin. A summary of the parameter values for the model is provided in *S1 Text, Section 2.2*.

Using the methods presented in *S1 Text, Section 1* with the parameter values listed in *S1 Text, Section 2.2*, numerical simulations and sensitivity analyses were conducted to examine the impact of various doses of dioxin exposure on hepatic and lipoprotein cholesterol homeostasis. The discrete sensitivity analysis, investigating ±10% changes in parameter values, did not reveal any concerns, as all parameter sensitivities with respect to steady-state concentrations were less than 1 in magnitude (see *S1 Text, Section 5*).

## 2.4. Anchor model of ovarian steroidogenesis

We set up a dynamic model reflecting steroidogenesis, as depicted in Fig 3. It can be found in *S1 Text* and GitHub.com/LBSA-VoitLab/TCDD_Chol. The system was formulated as an anchor model in the format of mass action, Michaelis-Menten, or Hill kinetics. The reactions were parameterized to match observed fold-changes in variables. The ODEs and parameter values of the model are listed in *S1 Text, Section 2.3*. The key model output (**Fig 7**) recapitulates the hormonal dynamics in the follicular and luteal phases semi-quantitatively [52,53].

The steroidogenic enzyme targets of dioxin vary with species and the state within the ovarian cells [31,32,36,38,54]. This fact renders it challenging to implement the inhibitory effect of dioxin in an enzyme-specific manner. To address this issue, we collectively assess the effect by modulating just the rate-limiting step of the steroidogenesis pathway, which is the conversion of cholesterol to P5 by the enzyme 20,22-desmolase (k_Des; Fig 3). This simplification is in line with research showing that the inhibitory effects of dioxin on P4, A4, T, and

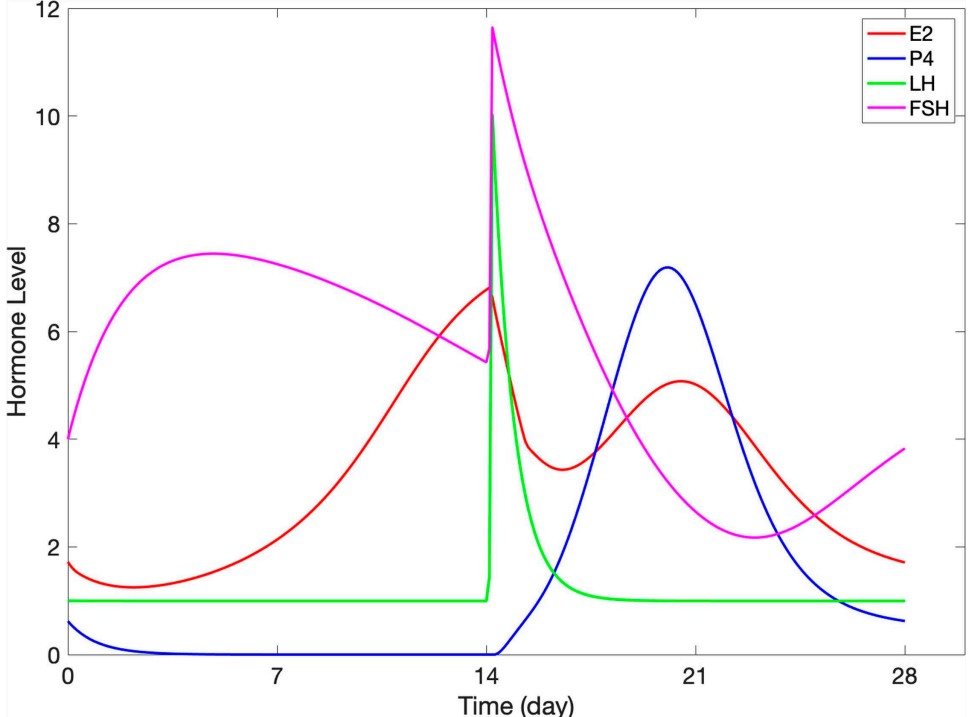

**Fig 7. Simulation results with the steroidogenesis anchor model.** The output captures the dynamics of gonadotropins (FSH and LH) and steroid hormones (E2 and P4) throughout one menstrual cycle.

E2 in mouse antral follicles can be reversed completely by oversupplying P5, suggesting that the main inhibition site of dioxin is upstream of P5 [32]. Thus, we simulate the effect of dioxin by altering k_Des to different fold-change levels relative to its default value. This linking of dioxin exposure and k_Des activity is furthermore supported by epidemiological information. Namely, a study [34] investigated the association between menstrual cycle length and serum dioxin level in young women who were exposed to dioxin caused by an industrial explosion. These women had not reached puberty at the time of the incident and had subsequently lived in the same area for 20 years. The study showed that for every 10-fold increase in serum dioxin concentration, the menstrual cycle was lengthened by 0.92 days. With the serum dioxin ranging between 10-10,000 part per trillion (ppt) in these subjects, the mean menstrual length varied by nearly 3 days.

Assuming that the increase in menstrual cycle length was caused primarily by an increase in the duration of the follicular phase, we implemented the dioxin dose-response by specifying a Hill function between dioxin concentration and the fold-change in k_Des activity so that high dioxin concentrations lowered the activity of k_Des to a value causing the follicular phase length to increase by 3 days.

To mimic epidemiological findings, representing distributions of health effects within populations, we ran Monte Carlo simulations with steroidogenesis model, where key parameter values were randomly sampled from pre-defined lognormal distributions (see below).

## 2.5. Interpersonal variability

Numerous studies have demonstrated substantial differences in the processes governing steroidogenesis (*e.g.*, [55–59]). Because these processes are characterized by different parameters and a complete assessment of all combinations is not really feasible, we resorted to the powerful method of Monte-Carlo simulation. In such a simulation, one value of each affected parameter is randomly selected from its known or assumed distribution. Using the combined, randomized set of all parameter values, the model equations are solved and all output measures of interest are stored. This step is repeated thousands of times, each time with a new set of randomized parameter values, and the overall result is a distribution of each of the chosen output measures.

## 3. Results

### 3.1. Anchor model of cholesterol biosynthesis

We previously examined the dioxin-mediated effects on hepatic and serum cholesterol homeostasis at various doses of exposure [13]. Additionally, we investigated the impact of varying levels of dietary cholesterol intake on the system.

The main result of this anchor analysis was that, despite a *decrease* in hepatic cholesterol synthesis due to dioxin exposure, the overall level of hepatic cholesterol *increased*. This at first surprising increase was shown to be attributable to increased transport of cholesterol from the plasma to the liver. Additionally, dietary cholesterol intake was demonstrated to have a more pronounced effect on hepatic cholesterol levels compared to plasma levels. Combined, dioxin exposure and a high-cholesterol diet were shown to lead to an even greater accumulation of hepatic cholesterol, highlighting the compounded impact of environmental toxins and dietary factors on liver cholesterol homeostasis.

Various simulation scenarios are possible with this anchor model. For instance, we may simulate different magnitudes of dioxin exposure, analyze the effects of varying dietary cholesterol intake on the system, and test how these variations, combined with dioxin exposure, impact hepatic and serum cholesterol levels [13]. As an illustration, we present in **Fig 8** the

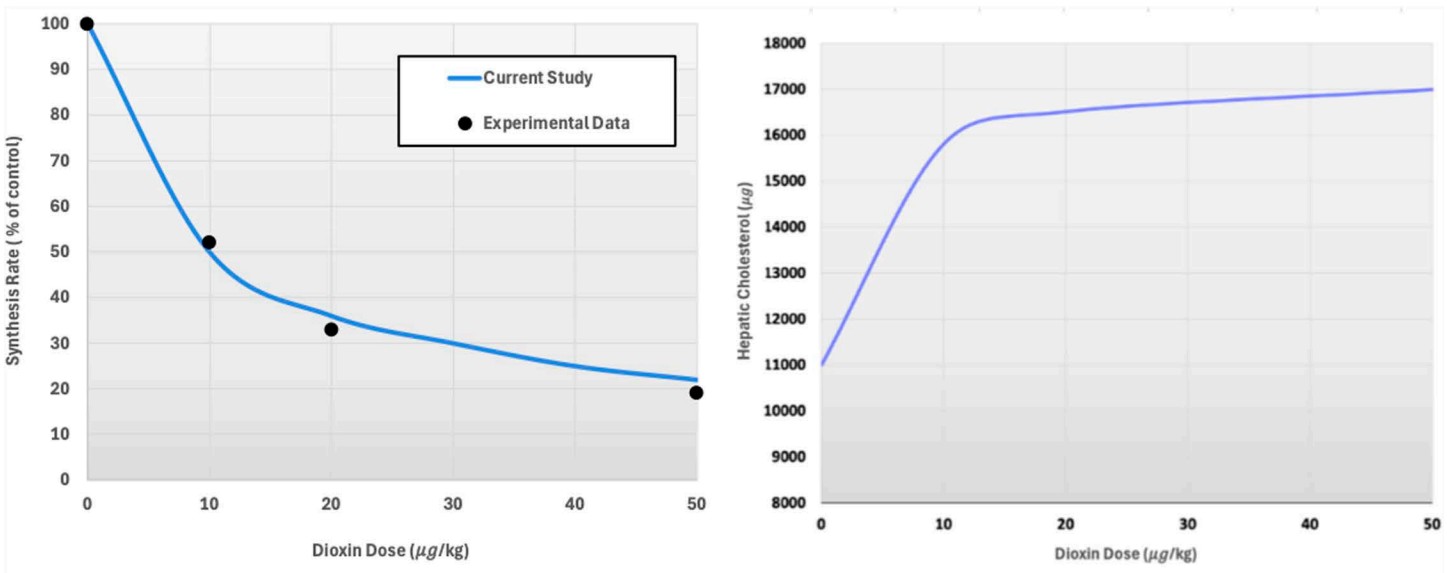

**Fig 8. Computational results from the biosynthesis anchor model (adapted from [ 13]).** Left Panel: The dose-response curve of the effect of dioxin on the biosynthesis rate of hepatic cholesterol is expressed as the percent of remaining activity in comparison with the control. The trendline was matched to fit experimental data (symbols) on mice [60,61]. Right Panel: The model results were formulated as a dose-response relationship characterizing the effect of dioxin on the steady-state amount of hepatic cholesterol.

result of one model analysis showing the effects of a series of dioxin concentrations on the cholesterol synthesis rate and the corresponding steady-state concentration of hepatic cholesterol. Interpolating these results yields a dose-response relationship between dioxin and the anchor's main output variable, hepatic cholesterol.

The amount of cholesterol produced under continuous exposure to any relevant dioxin dose of interest can be extracted from this dose-response relationship and will eventually be handed off to other variables of the template model. Specifically, the dose-response relationship will be used as a dioxin-dependent modulator of the healthy baseline flux (TCDD = 0) between the mevalonate entering the pathway and hepatic cholesterol level (see Fig 4).

## 3.2. Anchor model of cholesterol transport and metabolism in the plasma

**3.2.1. Effects of dioxin on cholesterol transport in plasma.** To evaluate the influence of dioxin on the lipoprotein transport system, we assessed the effects of varying doses of dioxin, spanning a range from 0 to 50 μg/kg body weight. Specifically, the assessment was executed by adjusting the parameters $DR_i$ for each dioxin-impacted flux (see *Methods*) such that the outcome matched literature information, which however was only available for a dose of 30 μg/kg [51]. We then calibrated the model using the previously developed dose-response curve for dioxin versus hepatic cholesterol [13].

In response to these dose-specific adjustments, the total plasma cholesterol, LDL, and HDL all decreased by roughly 25% following treatment with 30 μg/kg dioxin. At the same time, the gradual "loss" of plasma cholesterol with increasing dioxin levels turned out to be accompanied by strong increases in hepatic cholesterol, as they have been observed [13] (**Fig 9**).

Thus, similar to the dose-response curves previously reported for the cholesterol biosynthesis anchor model (Fig 8, [13]), the dioxin effect on hepatic cholesterol accumulation is noticeable even for low exposures and approaches saturation for high exposures. This

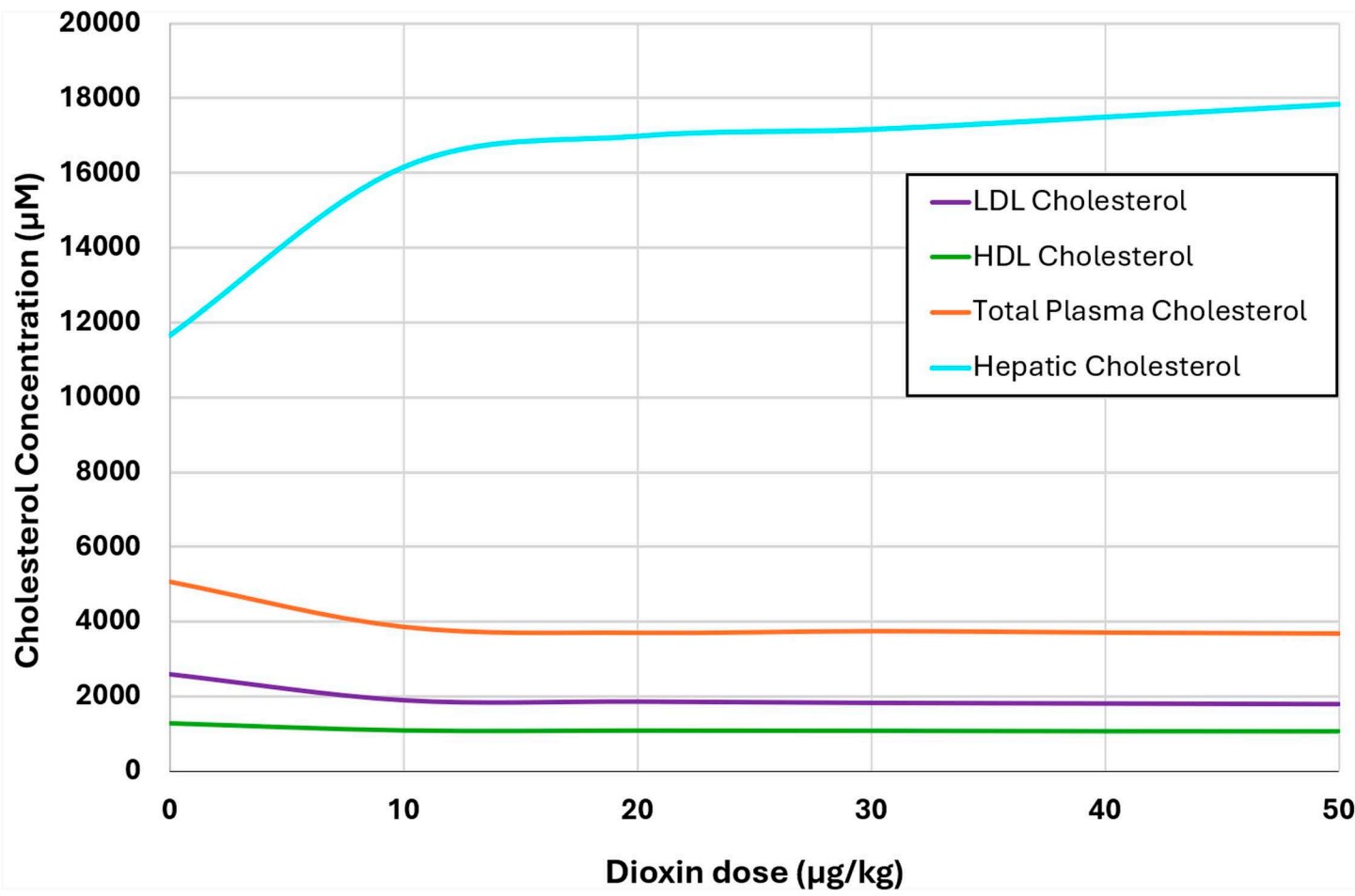

**Fig 9. Dose-response curves quantifying the effects of increasing dosage of dioxin on hepatic, and plasma LDL, HDL, and total cholesterol concentrations.**

increase in hepatic cholesterol under higher doses of dioxin is accompanied by a concomitant reduction in the concentrations of total plasma cholesterol, HDL, and LDL. Specifically, a dose of 30 μg/kg/day leads to an approximate 26% decrease of total plasma cholesterol, 17% decrease of HDL, and 30% decrease of LDL, which aligns well with observations in rodents [51].

**3.2.2. Effects of dietary cholesterol on LDL and HDL.** The model allows us to assess the potentially synergistic role of dietary cholesterol intake and its effects on hepatic and serum cholesterol. The recommended range of dietary cholesterol intake is approximately 200 mg/day, while high intake can reach 600 mg/day [62]. The model indicates that dietary cholesterol intake increases hepatic, VLDL, IDL, HDL, LDL, and total plasma cholesterol (**Fig 10**), which is consistent with literature reports [63].

Using these results, we explored with simulations the effects of combinations of dietary intake and dioxin exposure. Without dioxin exposure, the steady-state values of LDL and HDL cholesterol under a normal daily consumption of ~200 mg cholesterol are about 2,600 μM and 1,300 μM respectively. As expected, both LDL and HDL cholesterol levels increase with increasing dietary intake but show a decrease with increasing doses of dioxin (**Fig 11**). For the recommended cholesterol intake, the model predicts a decrease in LDL and HDL to 1,782 μM and 1,064 μM respectively, if the individual is exposed to a relatively high dose of 50

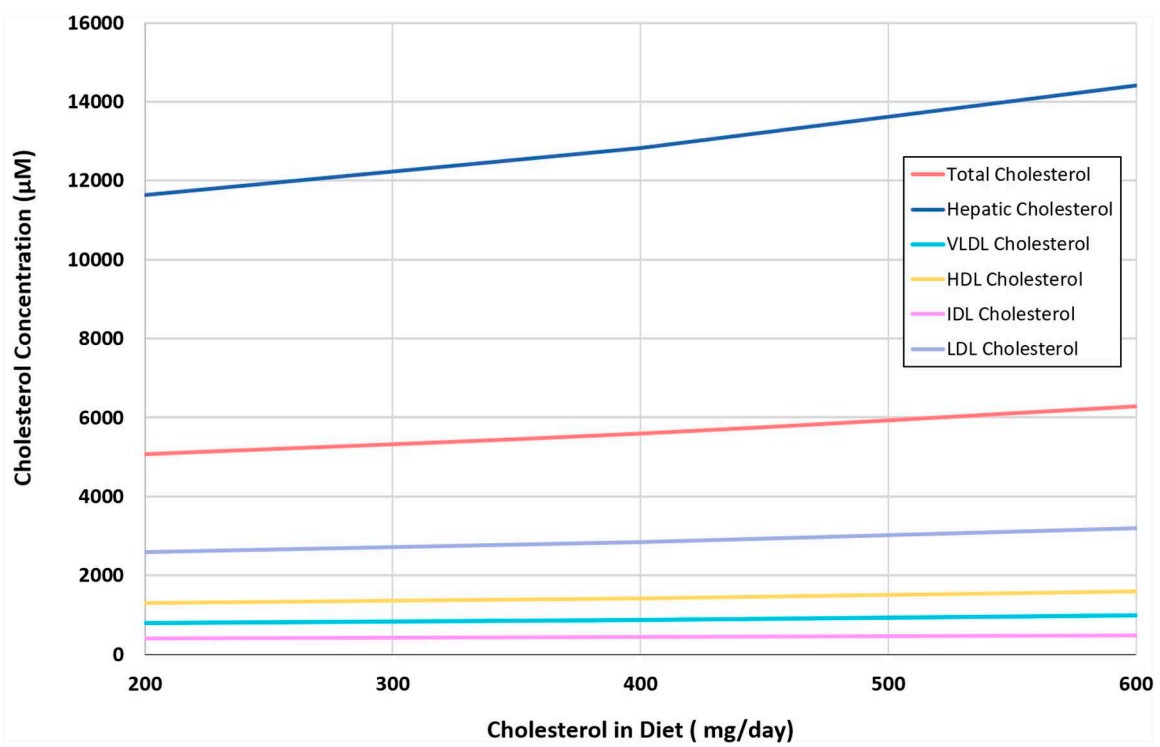

**Fig 10. Effect of increased dietary cholesterol intake on LDL, HDL, IDL, VLDL, as well as on hepatic and plasma cholesterol concentrations, according to the model.**

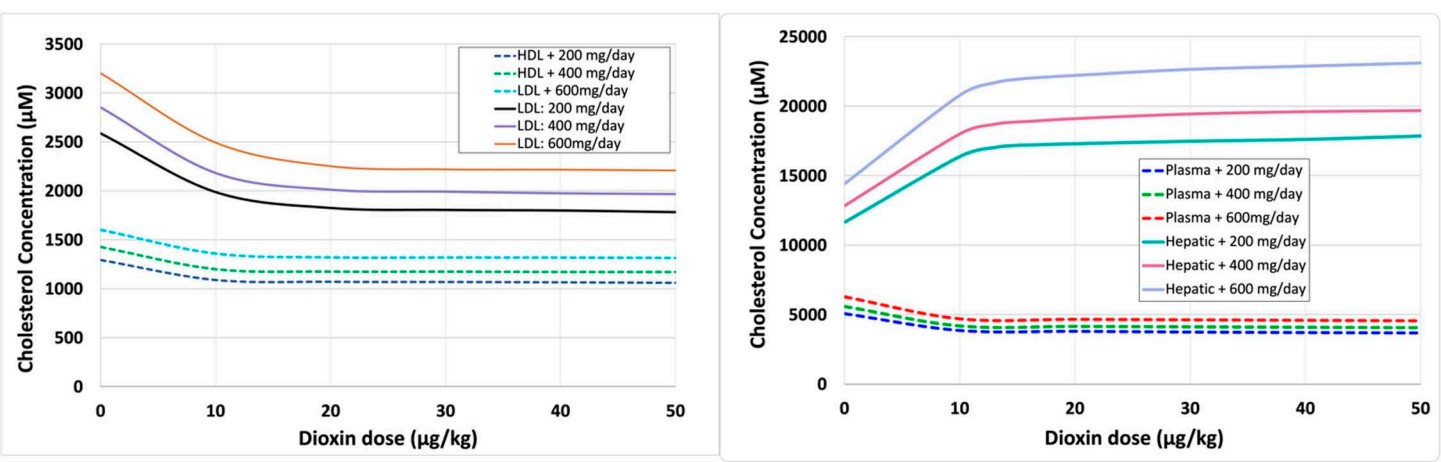

**Fig 11. Dose response curves accounting for the effects of dioxin and increased diet on the lipoprotein transport system.** Left: lipoproteins LDL and HDL. Right: Hepatic and plasma cholesterol. 200 mg/ day is the recommended amount of cholesterol in diet/day while 400 mg/day is high and 600 mg/day is considered very high.

μg/kg of dioxin. In comparison, for the same dose of dioxin, but with 600 mg/day of dietary cholesterol intake, the model predicts approximately 2,200 μM and 1,315 μM of LDL and HDL cholesterol respectively, which is close to the concentrations observed with the recommended dietary intake but with no dioxin exposure. Again, these decreases are accompanied by increases in hepatic cholesterol.

### 3.3. Anchor model of ovarian steroidogenesis

**3.3.1. Average model of the menstrual cycle.** Figs 3 and 7 depict the interplay of sex hormones over the course of a menstrual cycle, capturing the fluctuations of the key steroid hormones—E2 and P4—as well as the gonadotropins—FSH and LH. The model of this system was calibrated using literature information that benchmarks the average menstrual cycle at 28 days, with ovulation typically occurring around day 15 shortly after the LH surge. The model allows us to track the progression of E2, P4, FSH, and LH during the follicular phase from days 1 to 15, leading up to ovulation, and subsequently through the luteal phase from days 15 to 28.

**3.3.2. Population-based modeling of menstrual cycle variability.** The menstrual cycle length varies considerably within a population, due to both intra- and inter- individual differences [55,57–59], and particularly in the duration of the follicular phase [55,56]. Furthermore, there is a broad range of what constitutes "normal" hormonal levels across different individuals [52,53]. To address the documented variability of dioxin's effects on menstrual cycle duration and estradiol levels, we constructed a population model via Monte-Carlo simulation (see *Methods* 2.5), in which key parameters were randomized. These parameters quantify, respectively: the affinity constant reflecting the E2 threshold that triggers the LH surge ($J_{16}$), the clearance rates of estradiol ($k_{10}$) and progesterone ($k_{14}$) from the plasma; the rate of LH transport from pituitary into plasma ($k_{18}$), as well as the rate of antral follicle growth ($k_{21}$) (for further details, see *S1 Text, Section* 2.3). By this design, the simulation accounts for variations within a population and provides us with deeper insights into effects of dioxin on ovarian steroidogenesis.

The dynamic trends of E2, P4, FSH, and LH within the synthetic population are presented in **Fig 12** across the menstrual cycle. The simulation accounts for 100 individuals (grey). All trends retain their essential shapes but now exhibit variability. The figure shows individual trends (grey) as well as the means (solid colored lines) and 95% ranges (dashed lines).

**3.3.3. Impact of dioxin exposure on the average model of the menstrual cycle.** We evaluated the influence of dioxin on steroidogenesis by lowering the rate of the rate-limiting conversion of cholesterol to P5, which is catalyzed by 20,22-desmolase (k_Des; see *Methods*). Specifically, we subjected k_Des to different fold-change levels relative to the default value as indicated in **Fig 13**. For instance, a 20% reduction in k_Des corresponds to a commensurate 21% reduction in the peak E2 level, but it also delays the LH peak occurrence from the original day 14 to day 16.

**3.3.4. Effect of dioxin in the population model.** We quantified the population-wide impact of dioxin on steroidogenesis, again per Monte-Carlo simulation, for fold-decreases in k_Des. The results suggest that the lengths of the follicular phase exhibit a right-skewed distribution, as depicted in **Fig 14**, with the mean increasing as dioxin levels rise. Within the serum dioxin concentration range of 1 – 10,000 ppt, the follicular phase lengthens by approximately 3 days, starting from a mean of 14.4 ± 0.8 days to 17.3 ± 2.4 days, accompanied by increased variability, which includes instances of individuals displaying a 24-day follicular phase. The cumulative probability distribution of follicular phase lengths skews rightward as dioxin levels rise, suggesting that dioxin exposure impacts nearly the entire population under study.

### 3.4. Template model

**3.4.1. Overall Impact of Dioxin.** To quantify the impact of dioxin on cholesterol biosynthesis in the template model, we analyzed the effects of different doses of dioxin

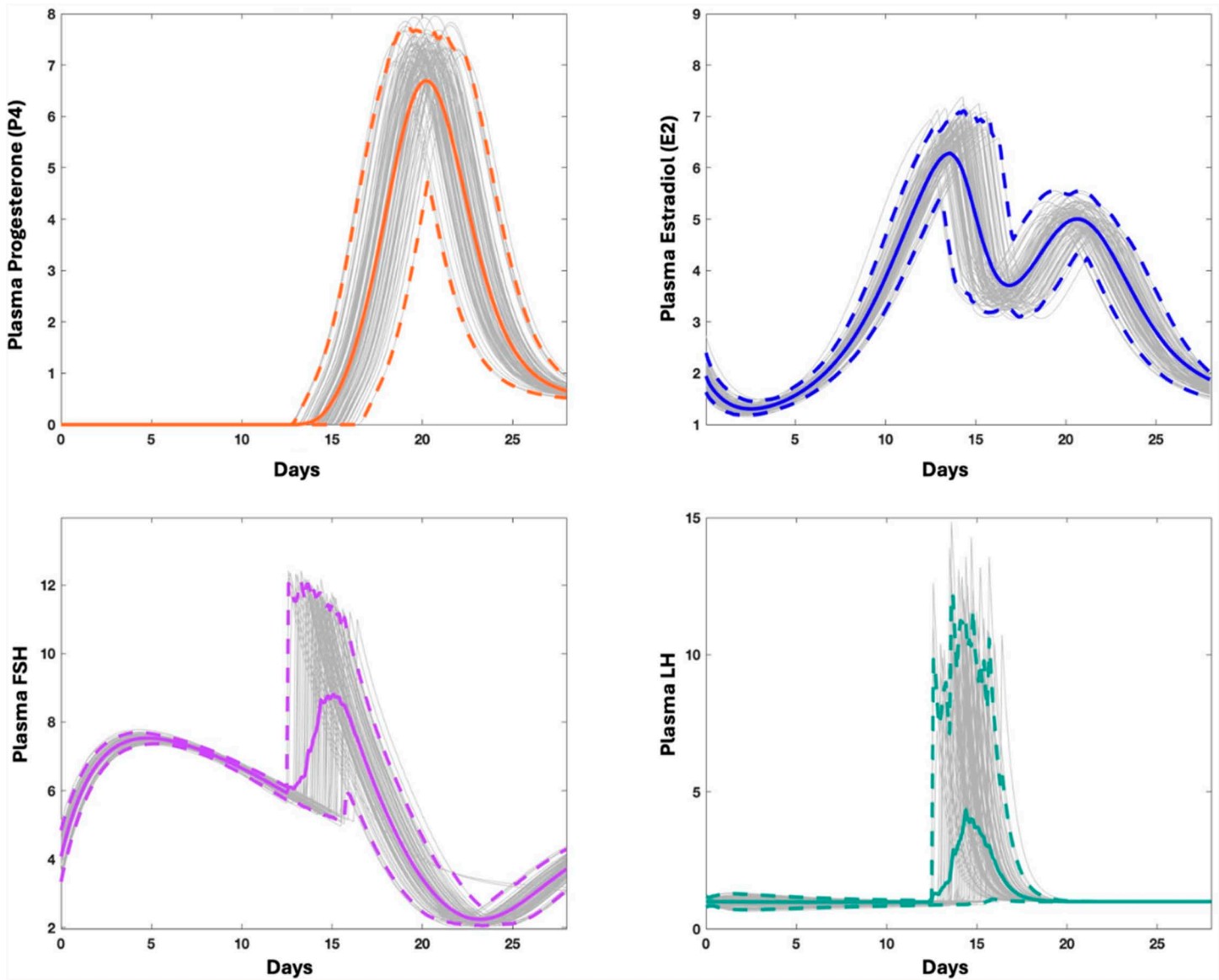

**Fig 12.  Dynamics of gonadotropins (FSH and LH) and steroid hormones (E2 and P4) throughout the menstrual cycle in a population of individuals.** Grey lines represent the data for 100 individuals. The solid lines denote the average values across these cases, while the dashed lines define the ranges within which 95% of the values lie.

ranging from 0 to 50 μg/kg body weight. To this end, the impact of dioxin on the overall rate of each anchor was incorporated into the template model through adjustments of the pertinent $DR_i$ quantities, as discussed in the *Methods* section and demonstrated before [13]. The $DR_i$ values were obtained from the dose-response curves of the anchor models. For instance, the alterations in rates of hepatic cholesterol biosynthesis were recorded and used to construct a dose-response (DR) relationship for the template model (**Fig 15**). Using this DR, we were able to predict the effects of dioxin on the various processes comprising the template model (Table 2).

With increasing doses of dioxin, the rate of cholesterol biosynthesis decreases [13], in line with what is documented in the literature (*e.g.*, [60]. The observed decline can be explained

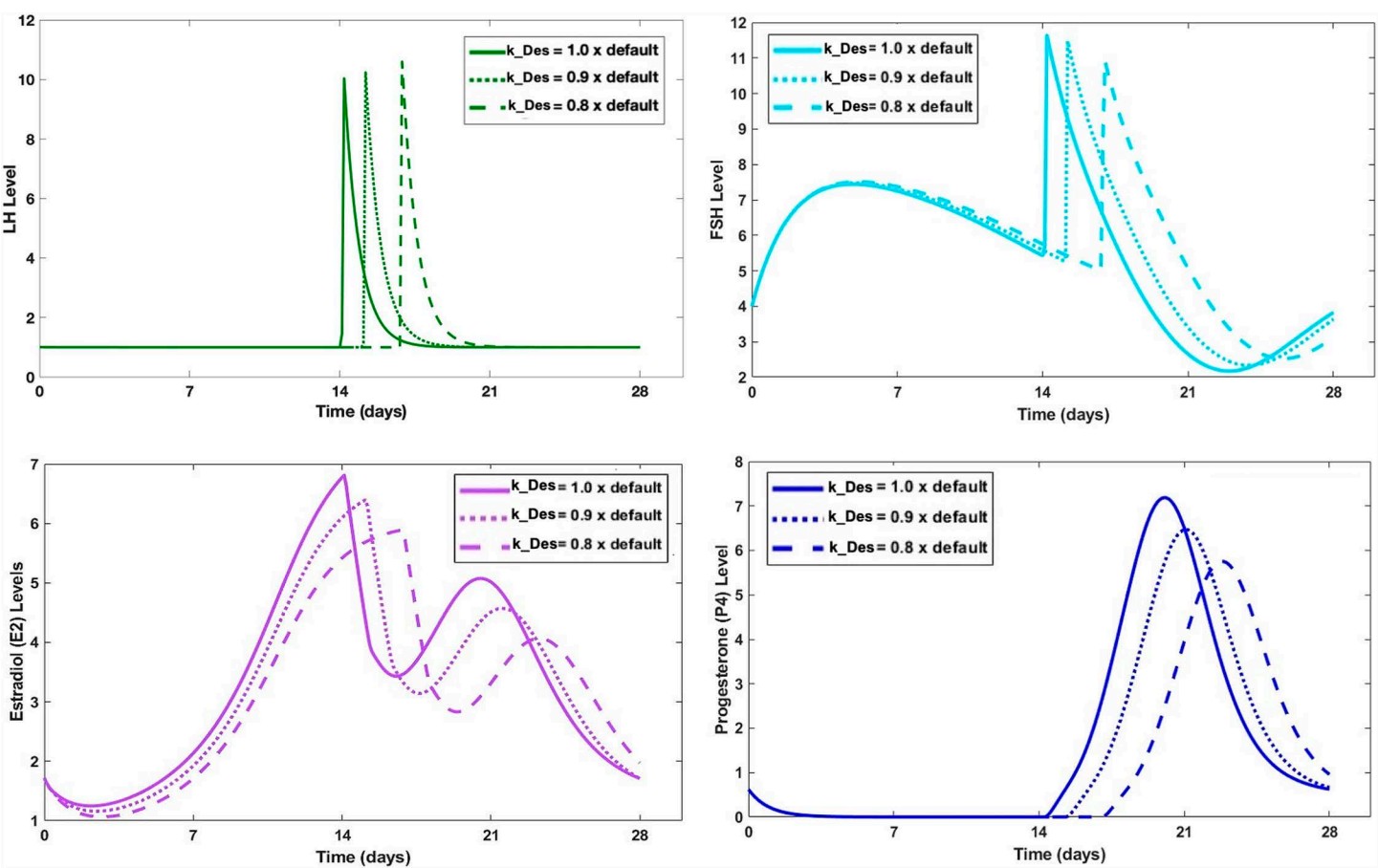

**Fig 13. Effects of varying k_Des activity (*i.e.*, conversion of cholesterol to P5 by 20,22-desmolase) on hormone dynamics based on the standardized menstrual cycle model.**

by the suppressed expression of multiple genes involved in biosynthesis. Increasing doses of dioxin are predicted by the template to raise peripheral cholesterol utilization and hepatic storage, while concurrently causing a reduction in plasma transport. These predictions are yet to be validated (or refuted) experimentally.

The decrease in biosynthesis is paralleled by a predicted decrease in sex hormone production. This model prediction aligns well with animal studies that indicate reduced estradiol production and with epidemiological studies associating lower estradiol levels with dioxin exposure in humans [34].

In contrast to biosynthesis and steroidogenesis, the model predicts an increase in the utilization of cholesterol in peripheral tissues. This predicted change is again consistent with the literature [64] and can also indirectly be related to human health, where such exposure has been linked to the development of ischemic heart disease [65], a condition precipitated by the accumulation of cholesterol within the coronary artery lining [66].

**3.4.2. Impact of diet on the template model.** The template model allows us to assess the role of dietary cholesterol intake and its interaction with dioxin exposure on a broader scale. We begin with the effect of dioxin at the baseline of TCDD = 0.

The template model indicates that an increase in cholesterol intake results in a decrease of cholesterol synthesis in the liver. A possible explanation is that the increase in hepatic

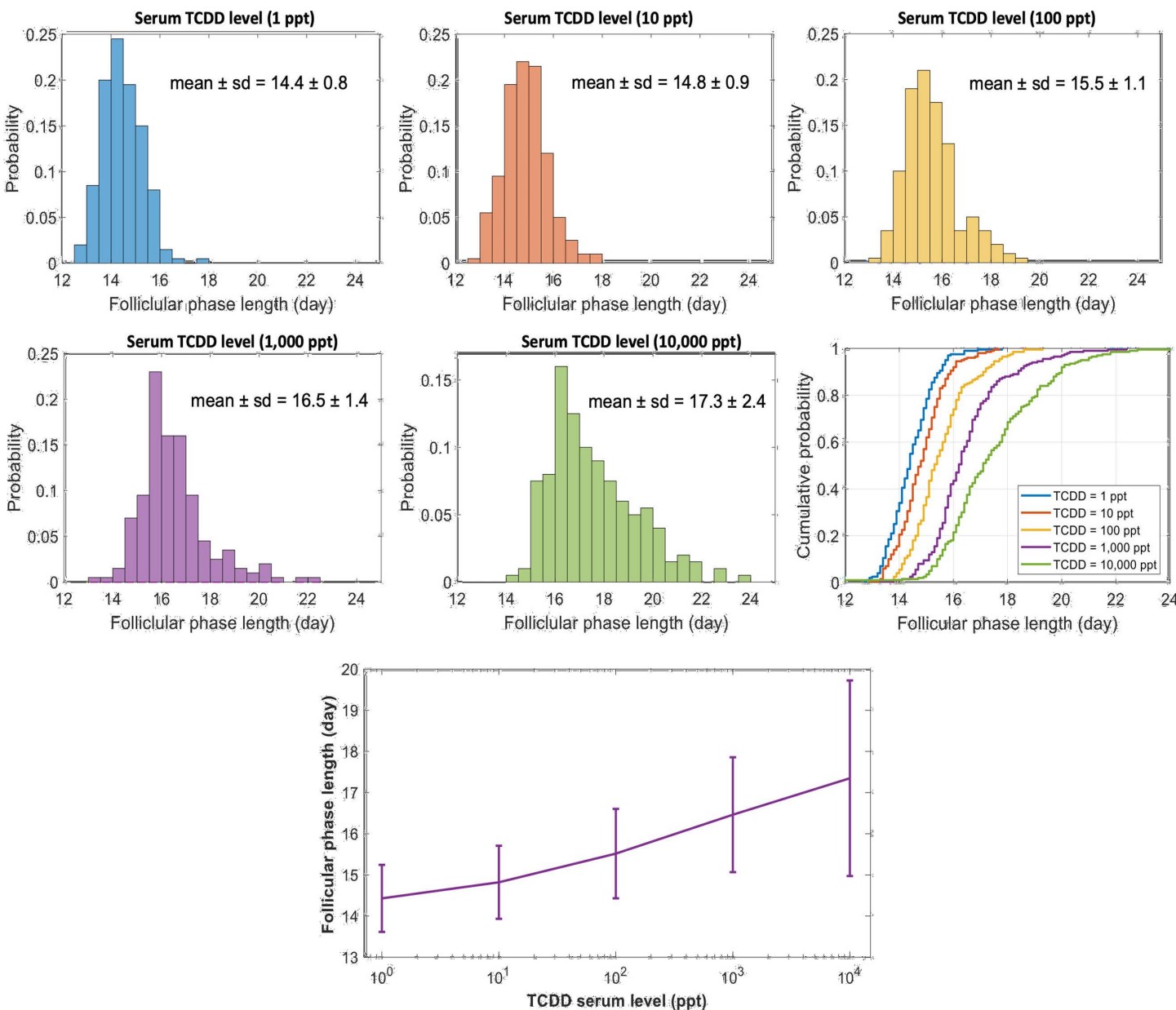

**Fig 14. Population-wide distributions of follicular phase lengths and corresponding dose-response curves.** Top panels: Frequency distributions and corresponding cumulative probabilities describing the relationship between dioxin serum levels (ppt) and follicular phase length. Bottom panel: Dose-response curve for a population, depicting the influence of dioxin serum level on the length of the follicular phase, expressed as the mean ± standard deviation.

cholesterol, as a result of increased dietary cholesterol intake, promotes the reduction of cholesterol synthesis enzymes through the down-regulation of SREBP [67]. Our simulations with the template model reveal a relationship between increased cholesterol in diet and decreased biosynthesis rate (Table 3), which is consistent with the results found in the cholesterol biosynthesis anchor model detailed in [13].

Overall, the template model indicates a decrease in cholesterol biosynthesis and plasma dynamics, accompanied by an increase in cholesterol storage and peripheral cholesterol utilization. At the same time, there is a significant decrease in steroid hormone production.

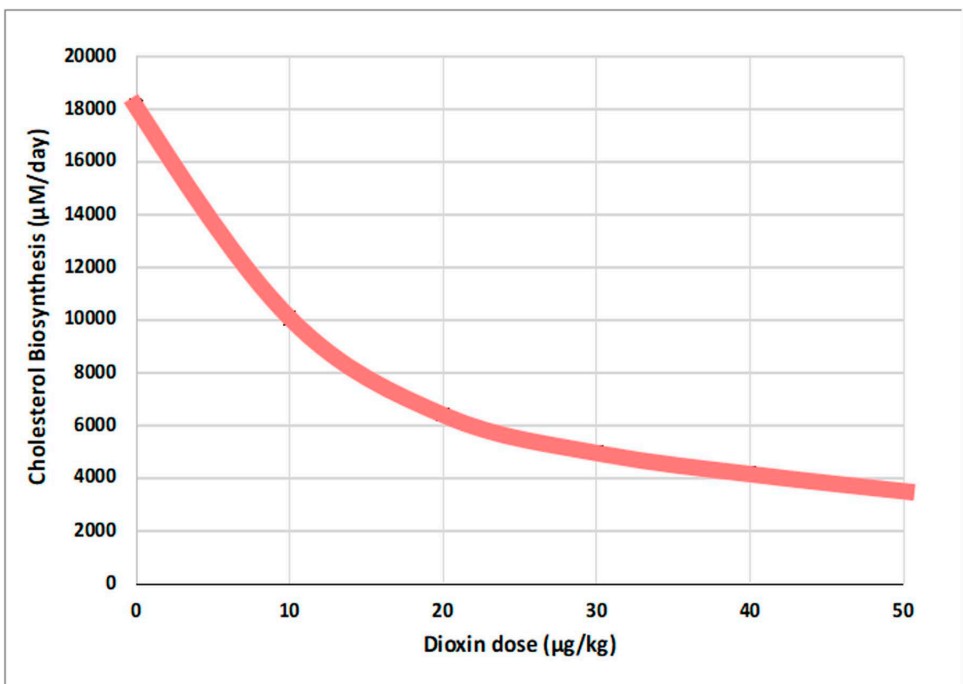

**Fig 15. Dose response curve of dioxin *versus* cholesterol biosynthesis, according to the template model.**

**Table 2. Dose-response relationships of the processes *versus* dioxin dose. Sections highlighted in grey represent simulation-based predictions generated by applying values of the dose-response curve for cholesterol biosynthesis (unshaded; Fig 8) to the entire template model, including hepatic cholesterol storage, peripheral tissue usage, plasma dynamics, and steroid hormone production.**

|  | Dioxin Dose | | | | | |
|---|---|---|---|---|---|---|
|  | 0 µg/kg | 10 µg/kg | 20 µg/kg | 30 µg/kg | 40 µg/kg | 50 µg/kg |
| Biosynthesis | 18150 | 10081 | 6424 | 5641 | 4196 | 3525 |
| Storage | 0.56 | 0.60 | 0.62 | 0.63 | 0.65 | 0.66 |
| Peripheral Tissue Utilization | 266 | 348 | 406 | 421 | 452 | 468 |
| Plasma Dynamics | 3266 | 2300 | 1663 | 1503 | 1180 | 1017 |
| Steroid Hormone Production | 1.34 | 0.94 | 0.68 | 0.62 | 0.48 | 0.42 |

**Table 3. Impact of diet on the various processes of the template model. The sections highlighted in grey represent simulation-based predictions generated by applying the dose-response curve for cholesterol biosynthesis (unshaded) to the template model.**

| Diet (mg) | Biosynthesis | Storage | Peripheral Tissue Utilization | Plasma Dynamics | Steroid Hormone Production |
|---|---|---|---|---|---|
| 200 | 18150 | 0.56 | 266 | 3266 | 1.34 |
| 400 | 15525 | 0.67 | 233 | 2859 | 1.66 |
| 600 | 14154 | 0.74 | 215 | 2634 | 1.87 |

A hand-waving explanation for this apparently counterintuitive decrease in flux rates in response to increased dietary intake is the availability of cholesterol from diet: The body does not need to produce as much cholesterol *de novo* if it receives it "for free" from the diet. This conclusion is in line with a documented link between heightened dietary cholesterol intake and the suppression of LDL receptor (LDLR) activity, which mediates the transport of cholesterol between the plasma and liver [68]. Specifically, the transport of LDL cholesterol to peripheral tissues has been found to be suppressed under rising cholesterol intake, likely due to the fact that the levels of cholesterol in those tissues are already increased [69]. Moreover, it has been found that heightened dietary cholesterol intake is correlated with an elevation of cholesterol in peripheral tissues [70]. It is postulated that this elevation may serve as a feedback inhibitor to cholesterol storage, thereby mitigating any further accumulation of cholesterol within these peripheral tissues. However, with strongly increased dietary cholesterol, a corresponding increase in cholesterol storage may still be observed [71].

**3.4.5. Combined impact of dietary cholesterol and dioxin exposure.** We used template simulations to explore the effects of combinations of dietary intake and dioxin exposure. Without dioxin exposure, the rate of hepatic cholesterol synthesis, under a normal daily consumption of ∼200 mg cholesterol, is about 18,150 μM/day (Table 4). As expected, the synthesis of hepatic cholesterol decreases with both increased dietary intake and increasing doses of dioxin, due to the accumulation of hepatic cholesterol (Fig 8). For the recommended cholesterol intake, the model predicts a decrease in the synthesis rate of cholesterol from 18,150 if an individual is exposed to no dioxin to about 3,525 μM/day, if exposure to high dose (∼50 μg/kg) of dioxin occurs. For the same dose of dioxin, but with 600 mg/day of dietary cholesterol intake, the model predicts a synthesis rate of approximately 2,683 μM/day (Table 4).

The sections highlighted in grey within Table 4 present simulation-based predictions generated by the simultaneous introduction of dioxin and increased dietary cholesterol intake.

**Table 4. Impact of diet and dioxin on the processes comprising the template model. Sections highlighted in grey represent simulation-based predictions generated by applying values of the dose-response curve for cholesterol biosynthesis (unshaded) to the template model.**

| | Dioxin Dose | | | | | |
|---|---|---|---|---|---|---|
| | 0 μg/kg | 10 μg/kg | 20 μg/kg | 30 μg/kg | 40 μg/kg | 50 μg/kg |
| **200 mg Cholesterol** | | | | | | |
| **Biosynthesis** | 18150 | 10081 | 6424 | 5641 | 4196 | 3525 |
| **Storage** | 0.56 | 0.60 | 0.62 | 0.63 | 0.65 | 0.66 |
| **Peripheral Tissue Utilization** | 266 | 348 | 406 | 421 | 452 | 468 |
| **Plasma Dynamics** | 3266 | 2300 | 1663 | 1503 | 1180 | 1017 |
| **Steroid Hormone Production** | 1.34 | 0.94 | 0.68 | 0.62 | 0.48 | 0.42 |
| **400 mg Cholesterol** | | | | | | |
| **Biosynthesis** | 15525 | 8572 | 5439 | 4771 | 3541 | 2970 |
| **Storage** | 0.67 | 0.71 | 0.74 | 0.74 | 0.76 | 0.77 |
| **Peripheral Tissue Utilization** | 233 | 305 | 356 | 369 | 396 | 410 |
| **Plasma Dynamics** | 2859 | 2016 | 1458 | 1318 | 1035 | 892 |
| **Steroid Hormone Production** | 1.66 | 1.17 | 0.84 | 0.76 | 0.60 | 0.52 |
| **600 mg Cholesterol** | | | | | | |
| **Biosynthesis** | 14154 | 7787 | 4928 | 4320 | 3201 | 2683 |
| **Storage** | 0.74 | 0.78 | 0.81 | 0.82 | 0.84 | 0.85 |
| **Peripheral Tissue Utilization** | 215 | 281 | 328 | 340 | 365 | 377 |
| **Plasma Dynamics** | 2634 | 1857 | 1342 | 1213 | 952 | 820 |
| **Steroid Hormone Production** | 1.87 | 1.32 | 0.95 | 0.86 | 0.68 | 0.58 |

Note that all results pertain to changes in processes and reveal no information regarding concentrations. To gain insight into changes in concentrations or amounts of cholesterol in different tissues, we need to explore the theoretical structure of T&A models.

## 4. Theoretical considerations regarding the structure and interpretation of T&A results

We have been using T&A models in an intuitive manner, adopting the notion of a template model as a high-level, coarse-grained umbrella structure and the anchors as models elucidating parts of the template in greater detail. At first glance, the mathematical structures of template and anchor models seem to be simple variations of the same theme. Closer inspection, however, reveals an interesting combination of differences and correspondences between template and anchor models in general. This combination is most easily seen for a dynamic system operating at a steady state.

### 4.1. Analogy with dual graphs

At the steady state, the structures of anchor and template models conceptually resemble "dual graphs" (*e.g.*, [72]). The topic of dual graphs has been discussed in the graph theory literature for a long time, using different terms, primarily "line graphs," but also "edge graphs" and "edge-to-vertex duals," among others. Line graphs were arguably first mentioned by Whitney in 1931/1932 [73]. A good review is the 1972 book by Harary and, in particular, the chapter on line graphs [74].

In the language of graph theory, the line graph G* of a graph G is constructed by exchanging edges and vertices. In our case, the model structures at steady state are directed graphs ("digraphs"), and the analogy in this situation is therefore called a "line digraph" [74,75]. For directed graphs, the vertex set of the line digraph G* corresponds to the edge set of G, and every directed edge in G* corresponds to an instance in G where the head of an edge $e_i$ meets the tail of another edge $e_j$. **Fig 16** shows a generic example.

In contrast to static graph theory, the literature on dual dynamic models is scattered, with rather different concepts using the term "dual" (*e.g.*, [76–78]). Closest in spirit to the situation

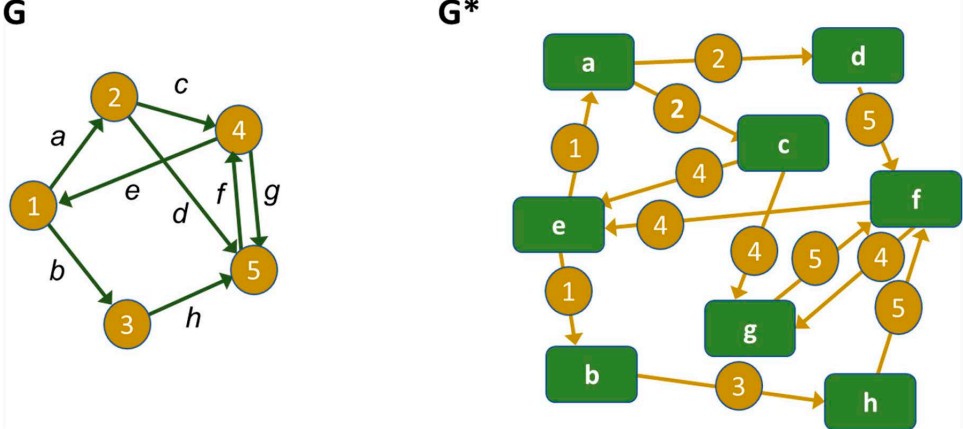

**Fig 16. Example of a directed line graph G and its (edge-to-vertex) dual graph G*.** The vertices of G* correspond directly to the edges in G, as indicated by letters a – h. The edges of G* correspond to vertices in G if one edge is entering this vertex and another one is leaving it. For instance, vertex ② in G is represented in the directed dual graph G* through the edges from ⓐ to ⓒ and from ⓐ to ⓓ, because these edges meet head to tail at vertex ② in G.

of T&A models is an article on flux duality [79], where the exchange of variables and fluxes in certain types of primal and dual differential equation models was investigated. One must emphasize at this point that the notion of line graphs or line digraphs is only an analogy to our situation of T&A models. Nevertheless, this analogy helps us set up anchors and templates in an appropriate manner, as described below.

A typical anchor model has the underlying graph structure of a typical metabolic or physiological model. For instance, the variables (vertices) of the cholesterol biosynthesis model (Fig 1) are metabolites, and the fluxes (directed edges) among them represent (possibly regulated) biochemical reactions, transport steps, or the formation or disassociation of complexes. This type of diagram represents the biosynthesis anchor well and, as a natural consequence of the typical model design applied here, there is perfect flux balance at any steady state: the fluxes into any of the variables collectively equal the fluxes leaving this variable in magnitude.

Surprisingly, these fundamental features are different in a template model. The variables (vertices) in a template model are usually physiological or biochemical *processes*, or conglomerations of such processes. In the case of the cholesterol template, for example, the variables are the mevalonate pathway, cholesterol storage, the various roles of (cholesterol-containing) plasma lipids, and so on, which are all processes altering the concentration of free or bound cholesterol. Concomitantly, the directed edges in the template model do not represent material fluxes in the traditional sense. Rather, they indicate modifying effects that may increase or decrease the activity of another template variable, which itself is a process. These effects either consist of handing molecules like cholesterol from one subsystem to another or they represent indirect modulations. Importantly, there is no material flow between template variables in a strict sense, because "Biosynthesis" does not become "Plasma Dynamics," in analogy to "Lanosterol" becoming "Cholesterol" in the anchor model. Instead, the variable "Dietary Intake" increases the amounts of cholesterol in the variables "Plasma Dynamics" and "Storage," and the variable "Biosynthesis" indirectly affects an increase in the input to the template variable "Sex Hormone Production." Furthermore, material may be generated or disappear within boxes (template variables). For instance, cholesterol disappears within the box "Sex Hormone Production," because it is converted into testosterone, estradiol, and other steroids (Fig 4). Within the template variable "Biosynthesis", cholesterol is apparently created out of nothing (Fig 4). Thus, a more accurate, although slightly awkward depiction of the cholesterol template is shown in **Fig 17**. It is of note that this representation actually runs counter to the standards of typical biological diagrams (Fig 6; [39]). For example, consider the signal emitted by a box like "Tissue Usage" on "Storage" within this diagram: It signifies that increased use of cholesterol in the peripheral tissues indirectly affects the storage of cholesterol in the liver, for instance by triggering its release from the storage compartment or by preventing available cholesterol from being stored.

The stark difference in model structures has implications for the stacking of sub-models during the zooming between templates and anchors. Specifically, it is possible that "anchors" associated with a template model can be converted into lower-level templates. This change comes with a change in model structure. As an example, "cholesterol storage in the liver" is currently an anchor model that contains as main variable the currently stored amount of cholesterol. However, if this anchor model is to become a template for an underlying model of finer granularity, the variables become the molecular and physiological processes of storing and releasing cholesterol in response to demands. The result could be a sub-template model, with its own anchors, for a lower organizational level.

Other examples of T&A-like models with processes as variables are discussed in *S1 Text, Section 6*.

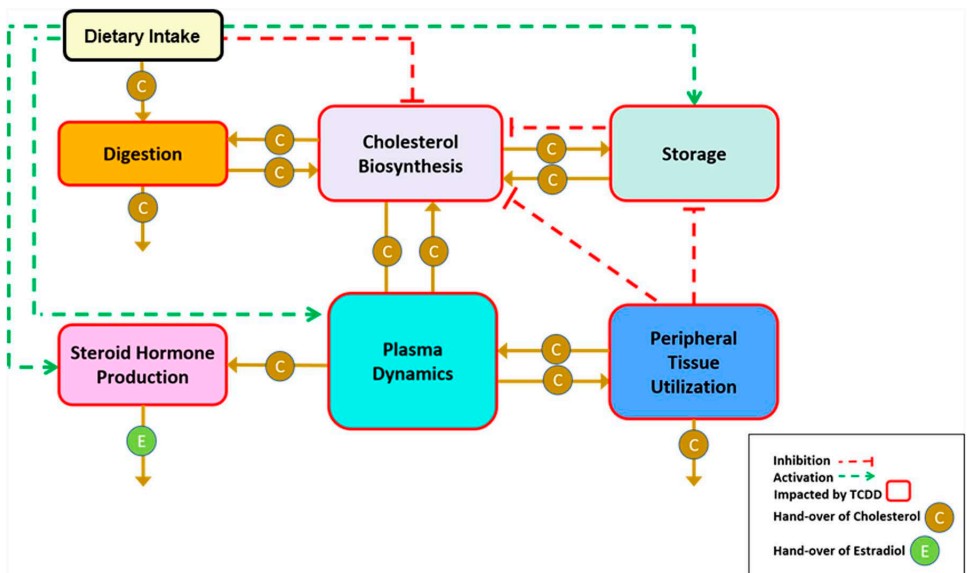

**Fig 17. Cholesterol template model as dual graph.** New representation of the cholesterol template model (Fig 4), where variables (anchors) are in truth processes and edges either represent the moving of cholesterol ⓒ among processes or indicate inhibitory signals - - -|. Note that the arrows differ from the standards of typical proper model diagrams (Fig 6).

### 4.1. From process rates to concentrations

All results of the T&A analysis (*e.g.*, Table 4) pertain to processes, as these are the variables of the template model. They do not say anything about concentrations or amounts of cholesterol in different organs or tissues. Somewhat of an analogy to this possibly perplexing aspect of the template analysis is the well-known method of flux balance analysis (FBA; [80]), which also focuses strictly on fluxes and their distribution under different conditions, while it does not shed light on concentrations. Similarly here, the template analysis reveals changes in the different processes of cholesterol handling. To obtain insight into the amounts of cholesterol, it is therefore necessary to do further model analysis.

As a simplified example, suppose we create a model of changes in the amount of hepatic cholesterol (HC), which for this illustration is drastically simplified (**Fig 18**).

The equation corresponding to the dynamics of HC, in generic terms, is:

$$\dot{HC} = k_D D + k_B B + k_{LD} LD + k_S \, fct(S, \, PU) - h_{LD} HC - h_S HC \tag{8}$$

Literature information is available to set an appropriate steady-state value for HC [49], and the rates of the processes of production and consumption, if unknown, can be adjusted to reflect this value. In most cases, a mass action model will suffice, but a default alternative could be a power-law function, for instance, for the cholesterol release from storage, which involves a signal.

Given this baseline model, each of the flux rates can be modulated by simulation results associated with dioxin exposure (Table 4), which renders it possible to study the effect of dioxin on the HC concentration. Indeed, it is possible to produce a dose-response curve for the effect of dioxin on HC, which can include different amounts of cholesterol intake through the diet.

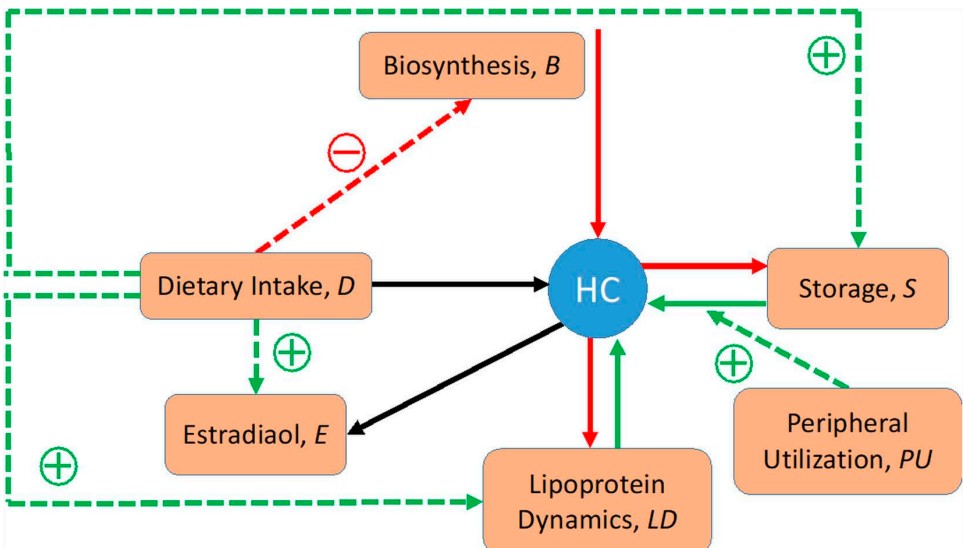

**Fig 18. Utilization of flux information (*e.g.*, from Table 4) to obtain information regarding cholesterol concentrations.** In this example, the concentration of interest is that of hepatic cholesterol.

## 5. Discussion

The purpose of this work was to gain a deeper understanding of the organism-wide health effects of the pollutant dioxin. Dioxin is known to affect pathways, transport proteins, storage and elimination processes, and developing an efficacious model therefore has to account for different levels of biomedical organization. As a strategy for addressing this task, we re-introduce the T&A framework, which was suggested twenty-five years ago for the purpose of inferring a set of common design concepts of posture and movement from different modes of locomotion among terrestrial species [11]. The authors pursued this goal by creating a model on a conceptual low-dimensional manifold describing the key aspects of locomotion. This invariant manifold was conceived as being embedded within a much higher dimensional space describing the various observed modes of locomotion, and all species-specific implementations were seen as intersecting in the subspace of the common design [11,12,81].

Here, we use the T&A framework not as a method for studying design features, but as an effective tool for addressing complex tasks in biomedicine and toxicology. As such, this tool may become a potentially very powerful NAM for dissecting multiscale modeling tasks into more manageable sub-tasks. NAMs have become the target of recent research efforts in toxicology [4,5].

We demonstrate that the T&A approach provides a systematic and intuitively organized strategy for multi-level analysis of complex biological systems. The template model serves as a high-level, coarse-grained representation that focuses on the main physiological components of a system and entails a relatively small set of variables and parameters. It reflects an overarching view of the entire problem space, thus providing an overall snapshot of the dynamic interactions among its core constituents. By contrast, the anchor models provide a more detailed and finely-grained view of specific biological details and mechanisms within the system. They allow the user to zoom into individual variables within the template, thereby permitting an account of intricate aspects of the underlying processes. Together, the template and anchor models form a hierarchical structure, allowing for both broad overviews and

in-depth analyses, thereby offering a versatile tool for multiscale modeling, while keeping computational cost down (see below).

The T&A strategy is directly applicable to the study of mechanisms with which dioxin alters the handling of cholesterol throughout the body. Thus, we designed and analyzed three detailed anchor models that were then used to inform the higher-level template model. The anchors capture the dynamics of hepatic cholesterol biosynthesis, lipoprotein transport, and the synthesis of sex hormones. The biosynthesis and lipoprotein anchors shed light on the impact of dioxin on hepatic and plasma cholesterol handling (including the roles of LDL, IDL, VLDL, and HDL cholesterol) and reveal an increase in hepatic cholesterol and a decrease in plasma cholesterol under dioxin exposure. The analysis specifically demonstrates that the upregulation of the SR-B1 and LDLR genes, which are involved in cholesterol transport between the plasma and liver, causes this decrease in plasma cholesterol concentrations and the concomitant accumulation of hepatic cholesterol. The steroidogenesis anchor demonstrates that a relatively high dose of dioxin exposure (serum concentration $=10^4$ ppt) leads to a reduction in estradiol synthesis, resulting in a lengthening of the menstrual cycle by nearly 3 days and consequently delaying ovulation. These findings align well with epidemiological studies [34]. Additionally, the model indicates a decrease in E2 levels with increasing doses of dioxin, which is again consistent with studies found in the literature [34].

In terms of the hierarchical structure of the T&A paradigm, the anchors correspond to the variables of the template model. Instead of carrying all details of the various anchor analyses forward, as it is attempted in many multiscale approaches, the T&A approach only uses key results from the anchors to inform the template. In our case of dioxin exposure, dose-response curves obtained with the anchors are used to modulate the variables in the template. As a result, the template offers, quasi in a minimalistic manner, a high-level overview of physiological processes and their alterations caused by external agents like dioxin. In doing so, the overarching template model captures the various processes that are explored in detail through the more fine-grained anchor models, which at this point include cholesterol biosynthesis, lipoprotein transport, peripheral cholesterol utilization, cholesterol storage, and steroid hormone production.

From a theoretical point of view, it is interesting to study the nature of variables in templates and anchors. While anchors by and large have the structure of conventional models, the variables of the template model represent processes, whereas the edges indicate how these processes in different tissues modify each other. The focus on fluxes in the template analysis is conceptually similar to approaches like flux-balance analysis [80], whose variables are exclusively flux rates. As a consequence, FBA does not make any statements regarding metabolite concentrations. Here we show how the information from flux distributions in the template can be used to gain insights into concentrations within a T&A model.

The T&A modeling approach carries distinct advantages in handling multiscale problems. In particular, the method enhances computational efficiency through its divide-and-conquer strategy and provides a flexible structure for model setup and the definition of variables across different levels. In contrast to other multiscale approaches, each anchor is separately analyzed, yielding a comprehensive record of essential input-output relationships that inform other anchors and calibrate the template model, while allowing for the interchangeability of anchor models without affecting the overall template structure.

Of course, the T&A approach is no panacea, and our dioxin exposure model, for example, is obviously simplified, like any other model. Even for a relatively straightforward anchor like the model of the mevalonate pathway, much more detail could—and maybe should—be considered, as it has been demonstrated that gene expression in the cholesterol biosynthetic

pathway differs among hepatocytes depending on where they are located within the lobules that form the microstructure of the liver [82–84]. The same is clearly true for other anchors, such as our model of steroidogenesis.

A possible further limitation is that we performed our analysis at the steady state of the pathway system, which may seem restrictive. However, one might expect that the pathway within the human body operates relatively close to the steady state and that the dose-response curves we determined with the anchor models are similarly applicable to situations not directly at, but close to the steady state. Somewhat related to the topic of dynamic-*versus*-steady-state operation is the question whether a separation of time scales is required between anchors and the template. Such a separation is here implicitly assumed, but it is unknown whether it is mathematically essential; this interesting question will require future investigation. Despite these questions and limitations, the proposed approach appears to be sufficient for revealing the essence of complex phenomena.

It goes without saying that the quality of any model, regardless of its sophistication, is fundamentally driven by the availability and accuracy of the input information used for its calibration. Furthermore, it is seldom possible to identify the mathematical structure of physiological processes directly from data. These caveats obviously pertain to our T&A analysis as well. With respect to input information, there are numerous ill-characterized uncertainties related to the values of parameters and the structure of the cholesterol handling systems. In particular, the scarcity of human data and our substitution with mouse data represent a significant caveat. This aspect mandates caution with interpretations because, for instance, some responses of serum cholesterol levels to dioxin appear to vary between humans and mice [51,85,86].

To address these differences in circulating plasma cholesterol levels, while concurrently accounting for the accumulation of cholesterol in the human liver and integrating into the model additional pathways of dioxin's impact on organs besides the liver, it would be desirable to develop separate human-only or mouse-only models. Alas, the available information is currently insufficient to develop such targeted models. A critical advantage is that the models proposed here can easily be adjusted as new data become available, including the substitution of anchors with alternate models.

Regarding the "true" mathematical structure of the model, we resorted to the use of mass-action and power-law representations. These formulations are guaranteed by Taylor's approximation theory to be accurate at least close to an operating point, if appropriate parameter values are used. Thus, while true functions will probably remain elusive for a very long time, the power-law-based differential equations in the model arguably constitute the most straightforward and least biased alternative (see [39] as well as references therein and above).

While certainly not complete or perfect, the T&A approach was shown to be sufficient for revealing the essence of the complex phenomenon of dioxin affecting cholesterol handling. In particular, the template model appears to perform well in capturing system-wide effects of dioxin, and as far as experimental or clinical information is available, the model results align well with reports in the literature. As an example, the template demonstrates a dose-dependent reduction in cholesterol synthesis, concurrent with a reduction in estradiol production. At the same time, cholesterol storage and utilization in peripheral tissues are increased. Furthermore, as the combined biosynthesis-plasma anchor model shows, the result of dioxin exposure is an overall increase of cholesterol in the liver, leading, in extreme cases, to non-alcoholic fatty liver disease [13].

Given the novelty of the T&A modeling paradigm, as it is proposed here, one must realize that this approach is still developing. While the T&A model appears to function properly for our application, it is obviously simplified, like any other model. Clearly, further complexity

could be added to the anchor models, for instance, to account for the intricacies of cholesterol absorption from diet and the formation of bile from hepatic cholesterol, which involves a large number of enzymes and transport proteins in a multi-step process [87]. Similarly, the anchor model of steroidogenesis examines the effects of dioxin during the follicular phase to help us understand its impact on menstrual cycle duration. Although it is noted that most variability of cycle length is usually derived from varying lengths of the follicular phase [88], the model is still woefully simplified, for example, considering that the corpus luteum also synthesizes steroid hormones similarly to the antral follicles during the follicular phase, and thus could potentially be influenced by dioxin exposure in a secondary manner. As more information becomes available, this and other anchor models may be refined, thereby enhancing their accuracy and predictive capability.

It is evident that much remains to be done, but it also becoming clear that the adaptation of the T&A approach proposed here offers substantial advantages without unduly losing resolution. The main appeal of the method is that the very complex task of multiscale analysis can be split into several, almost independent subtasks. This divide-and-conquer strategy directly enhances clarity for the model design phase, as well as computational efficiency. Unlike other methods, each anchor is analyzed individually, and while each anchor analysis generates numerous results, only key results, such as input-output relationships and their susceptibility to dioxin exposure are retained as dose-response curves; they could even be stored as computationally efficient, tabular results. Only these key results are used to inform the template model. One might add that anchor models can be replaced with alternative representations without affecting the structure of the template or other anchors. Finally, the T&A approach provides flexible guidelines for the setup of the two model types and for defining variables and interactions across scales. These guidelines may pertain to average models for populations or may be applied to customized, individually personalized models in straightforward extensions [89–91], These personalized models could, in turn, become the basis for risk assessments and virtual clinical trials [92,93].

Multiscale approaches are clearly important for biomedical modeling, and the revisited T&A paradigm provides a novel, effective tool for complex modeling tasks as it provides a comprehensive and holistic perspective that accommodates a range of organizational scales. Furthermore, an approach like T&A can readily be tailored toward many different research goals and applications, which could, for instance, support traditionally difficult toxicological extrapolations of *in vitro* to *in vivo* results [5] and the important translation of observations between different species [94]. For our example of dioxin exposure, this translation poses an interesting challenge, as some enzyme activities and the crucial dioxin-AhR mechanism are species-specific [95,96].

As one future application, the T&A framework, as a NAM [4], holds potential for advancing personalized risk assessments and personalized medicine [89–91] in general. It could also become relevant and effective for enhancing drug discovery and development. The current version of the model is mostly calibrated with average data, but we also indicate how variability may be taken into account. We illustrate this aspect with the example of inter-individual variability in menstrual cycle length within a population, where the production of estradiol is affected by dioxin. For future uses in personalized medicine, the model design is readily customizable, according to published personalization methods [89–91,97,98] that are based on individualized adjustments of relevant parameters and thus on personal data. These examples render it evident that the T&A platform is versatile and can be expanded in various directions to simulate the body's response to a wide array of pharmaceutical and toxic compounds, thereby potentially serving as a predictive tool for assessing the behaviors and interactions of drugs and environmental exposures within the body [99].

In conclusion, the T&A framework holds promise for multiscale modeling, for a deeper understanding of complex systems, and for advancing personalized medicine and risk assessment, interspecies translation [94], and drug testing through virtual clinical trials [100,101].

## Supporting information

**S1 Text.**   Fig A. Sensitivities of plasma cholesterol, LDL, and HDL. All relative sensitivities are below 1 in magnitude. Fig B. Results of a local sensitivity analysis for steroidogenesis pathway anchor model. All parameters have a sensitivity coefficient below 1 in magnitude. Fig C. Results of a sensitivity analysis of the template model. All parameters values exhibit changes in the steady-state values of less than 10% in magnitude, corresponding to absolute values of relative sensitivities <1. Fig D. Generic PBPK model as a multiscale structure. The primary scale of granularity reflects organs and tissues, but the drug concentrations in these organs are subject to molecular and cellular events, such as sequestration and degradation in the liver. In this illustration, the drug or toxicant enters the body per inhalation.
(DOCX)

## Acknowledgments

The authors would like to thank Dr. Shuo Xiao for helpful technical discussions regarding steroidogenesis and the menstrual cycle.

## Author contributions

**Conceptualization:** Carla M. Kumbale, Qiang Zhang, Eberhard O. Voit.

**Data curation:** Carla M. Kumbale.

**Formal analysis:** Carla M. Kumbale, Qiang Zhang, Eberhard O. Voit.

**Funding acquisition:** Qiang Zhang, Eberhard O. Voit.

**Investigation:** Carla M. Kumbale, Qiang Zhang, Eberhard O. Voit.

**Methodology:** Carla M. Kumbale, Qiang Zhang, Eberhard O. Voit.

**Project administration:** Eberhard O. Voit.

**Software:** Carla M. Kumbale, Qiang Zhang, Eberhard O. Voit.

**Supervision:** Qiang Zhang, Eberhard O. Voit.

**Validation:** Carla M. Kumbale, Qiang Zhang.

**Visualization:** Carla M. Kumbale, Qiang Zhang, Eberhard O. Voit.

**Writing – original draft:** Carla M. Kumbale, Qiang Zhang, Eberhard O. Voit.

**Writing – review & editing:** Carla M. Kumbale, Qiang Zhang, Eberhard O. Voit.

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
