## [Decision Letter · Decision Letter 0]

28 Feb 2024

Dear Dr. Voit,

Thank you very much for submitting your manuscript "Multiscale Biomedical Systems Analysis through Template-and-Anchor Modeling" for consideration at PLOS Computational Biology.

As with all papers reviewed by the journal, your manuscript was reviewed by members of the editorial board and by several independent reviewers. In light of the reviews (below this email), we would like to invite the resubmission of a significantly-revised version that takes into account the reviewers' comments.

We cannot make any decision about publication until we have seen the revised manuscript and your response to the reviewers' comments. Your revised manuscript is also likely to be sent to reviewers for further evaluation.

Sincerely,

Andrew D. McCulloch, Ph.D.

Academic Editor

PLOS Computational Biology

Pedro Mendes

Section Editor

PLOS Computational Biology

Reviewer's Responses to Questions

**Comments to the Authors:**

Reviewer #1: Kumbale et al. propose adapting the Template and Anchor Modeling methodology, originally designed for biomechanics models, to facilitate multilevel modeling from molecular networks to tissue/organism levels.

The manuscript presents a complex set of ideas with potential, but several questions arose as I finished reading it, mainly regarding clarity in exposition and organization.

While the paper introduces the Template and Anchor (T&A) approach for multiscale modeling, the conceptual framework of T&A modeling remains somewhat unclear. The distinction between template and anchor models seems inconsistently articulated throughout the paper, potentially causing confusion for both myself and readers unfamiliar with the concept. A clearer definition and differentiation between template and anchor models in the introduction section could help, but may not sufficiently elucidate the adaptation from biomechanics to biomolecular levels. The transition from a mechanistic-like model to a graph-theoretical model needs clearer explanation. The current structure of the explanation gives the impression of a two-level modeling approach, rather than a multilevel one. Did I misunderstand this? For instance, how would one use the methodology to progress from molecular networks to tissues, organisms, and populations/epidemiological groups? How would more than two levels be addressed?

While the paper discusses practical applications of the T&A approach, such as personalized medicine and translational research, it lacks specificity in describing how the approach can be applied in real-world scenarios. While the example presented throughout the paper is helpful, it is insufficient to fully illustrate the complex concepts. There are two way I can see to improve this. One is by creating a set of sections or a detailed methodology (or a second accompanying manuscript) where the concepts are properly formalized. The alternative, is providing more examples with similar depth to the one you use to enhance readers' understanding of the underlying concepts. Highlight specific challenges addressed and insights gained through T&A modeling in diverse contexts.

Additionally, while the paper acknowledges potential limitations and challenges associated with T&A modeling, the discussion remains relatively brief and lacks depth. Expanding the discussion section to include a more comprehensive analysis of the limitations and challenges of T&A modeling would be beneficial. This could be done by including additional examples that are unrelated to the system that is used throughout the paper.

The paper emphasizes the importance of empirical validation for T&A models but falls short of providing concrete guidance on validation methodologies and best practices. Utilizing more than one example to illustrate the framework and validate its models may facilitate this. The paper could also benefit from a more comprehensive analysis of potential avenues for advancing T&A modeling in biomedical research.

There are some additional comments and typo that you can find in the attached pdf.

Reviewer #2: The manuscript revisits the Template-and-Anchor (T&A) modeling framework, originally proposed for design concepts from various manifestations of biological locomotion. The manuscript emphasizes the T&A framework's utility in simplifying complex multiscale biomedical models by separating them into high-level template models and more detailed anchor models. The template model describes overarching bioprocess components with minimal variables, while anchor models delve into specific biological details of subsystems. This separation enhances computational efficiency, provides a structured approach to model development, and maintains flexibility by allowing anchor models to be independently updated or replaced. The manuscript illustrates this framework's application in the modeling of dioxin exposure and its effects on cholesterol dynamics, showcasing its potential.

The manuscript's exploration of the Template-and-Anchor (T&A) framework, particularly its application to biological systems modeling, is commendable. The presentation of the T&A framework is insightful, but its application to dioxin exposure needs clarifications (see Major Comments). I believe this work will likely be well-received by practitioners of biosystems modeling.

Major comments:

1. The manuscript provides a general description of the Template-and-Anchor (T&A) framework. Unfortunately, it lacks detailed implementation specifics, particularly in the dioxin exposure case study. Essential elements such as the interaction between template and anchor models, specific equations governing the dioxin model, and the nature of information exchanged between template and anchor are notably absent. This omission hampers the reader's ability to fully understand and appreciate the model's complexity and operational mechanisms. For example, it is not clear how one would get the simulation result in Fig. 4 since no information of the model that was solved was provided.

2. Despite the mention of future code availability on GitHub, the current lack of access to the model's code restricts the ability to replicate the study and/or delve into the technical intricacies of the framework. The absence of code, coupled with the lack of mathematical detail and algorithmic descriptions in the manuscript, significantly detracts from the research's reproducibility, thus limiting its utility and impact within the scientific community.

3. The similarity between the Template-and-Anchor (T&A) framework and block diagrams in process control, particularly in how both describe input-output relationships through modular components, deserves a comparison. Like T&A template, block diagrams use blocks to represent the relationships between variables, where such relationships are modeled using transfer functions. The anchors could be seen as complex versions of block diagram transfer functions. A more explicit comparison highlighting the T&A framework's unique contributions and advantages in modeling biological systems, beyond what traditional block diagrams offer, would greatly enhance the manuscript's impact.

**Have the authors made all data and (if applicable) computational code underlying the findings in their manuscript fully available?**

Reviewer #1: **No: ** Not yet. they say they will provide the code on GitHUb. I would not accept the paper before a link to the repository is provided.

Reviewer #2: **No: ** The code generating the model simulations was not provided with the manuscript for review.

PLOS authors have the option to publish the peer review history of their article (what does this mean? ). If published, this will include your full peer review and any attached files.

**Do you want your identity to be public for this peer review?** For information about this choice, including consent withdrawal, please see our Privacy Policy .

Reviewer #1: No

Reviewer #2: No
---

## [Editor Report · Decision Letter 1]

12 Aug 2024

Dear Dr. Voit,

Thank you very much for submitting your revised manuscript "Multiscale Biomedical Systems Analysis through Template-and-Anchor Modeling" for consideration at PLOS Computational Biology. As with all papers reviewed by the journal, your manuscript was reviewed by members of the editorial board.

You opted to split your text into two separate articles which raises several issues. Individual articles must be assessed independently and thus would need independent submissions (and these would merit their own decisions independently too). In this case it seems to us that neither of these articles would be sufficient on their own. Part 1 would lead to requests for an illustrative example (as already happened in the reviews of your original submission), while the Part 2 would raise questions about the methodology not being clear.  If the two are so interwoven that they require joint review, then they would better form a single article. Note that PLOS Computational Biology does not have a limit on article size.

At this point we will need you to chose your preferred option: 1) integrate these two articles into a single one and continue the revision process, or 2) address the previous reviewers' comments for the first manuscript and later submit part 2 as a separate manuscript that refers to the first as a normal citation (of course you would need to be sure of acceptance of the first before submitting the second).

In either case when you re-submit your revised version, please do it in the context of the previous reviewer comments and include a reply to them just as you did now. We will then send that version to the reviewers. If you prefer to communicate with editors by email, send it to <ploscompbiol@plos.org>, including the manuscript number in the subject line and requesting at the top of the message body that it be passed on to the editors.

Sincerely,

Pedro Mendes, PhD

Section Editor

PLOS Computational Biology

---

## [Decision Letter · Decision Letter 2]

7 Nov 2024

PCOMPBIOL-D-24-00128R2Analysis of Systemic Effects of Dioxin on Human Health through Template-and-Anchor ModelingPLOS Computational Biology Dear Dr. Voit, Thank you for submitting your manuscript to PLOS Computational Biology. After careful consideration, we feel that it has merit but does not fully meet PLOS Computational Biology's publication criteria as it currently stands. Therefore, we invite you to submit a revised version of the manuscript that addresses the points raised during the review process. Please submit your revised manuscript within 30 days Jan 07 2025 11:59PM. If you will need more time than this to complete your revisions, please reply to this message or contact the journal office at ploscompbiol@plos.org. Please include the following items when submitting your revised manuscript:* A rebuttal letter that responds to each point raised by the editor and reviewer(s). You should upload this letter as a separate file labeled 'Response to Reviewers'. This file does not need to include responses to formatting updates and technical items listed in the 'Journal Requirements' section below.* A marked-up copy of your manuscript that highlights changes made to the original version. You should upload this as a separate file labeled 'Revised Manuscript with Track Changes'.* An unmarked version of your revised paper without tracked changes. You should upload this as a separate file labeled 'Manuscript'. If you would like to make changes to your financial disclosure, competing interests statement, or data availability statement, please make these updates within the submission form at the time of resubmission. Guidelines for resubmitting your figure files are available below the reviewer comments at the end of this letter. We look forward to receiving your revised manuscript. Kind regards, Pedro MendesSection EditorPLOS Computational Biology

Feilim Mac Gabhann

Editor-in-Chief

PLOS Computational Biology

Jason Papin

Editor-in-Chief

PLOS Computational Biology

 **Journal Requirements:** **Additional Editor Comments (if provided):** Please try to follow the reviewers suggestions; most likely the revised version may not need further reviewing if you can follow those suggestions.**Reviewers' comments:** Reviewer's Responses to Questions

**Comments to the Authors:**

Reviewer #1: This paper continues to improve in clarity and quality, and the T & A methodology shows considerable potential. However, there are still some areas that require further clarification. Below, I provide general comments, while detailed feedback is included in the marked-up PDF file of the manuscript.

Bibliography: The references need to be revised. The current numbering starts with [2], while [1] only appears after [13]. Although [1] is cited in the abstract, it is my understanding that references should not be included in the abstract. If I am mistaken, I apologize, and the numbering is correct as it stands.

Symbolic Representation of Template and Anchor Models: Conceptually, using the same symbolic notation for both the Template and Anchor models creates confusion and complicates the understanding of the methodology. While I do not have a definitive solution, I suggest two possible alternatives:

a) Use the conventional representation for Anchor models, where state variables are represented as typical components, and processes are indicated by flux arrows. Then, adopt a modified graph-theoretical representation (such as those shown in Figures 16 or 17), but ensure that symbols or arrow types are distinct from the flux arrows used in the Anchor models.

b) Alternatively, employ a simple graphical notation similar to the Systems Biology Graphical Notation (SBGN) for the Template model.

In my view, this is the primary methodological issue that needs improvement to enhance the overall clarity of the approach.

Parallel Analysis for Anchor Models: It would be beneficial to conduct parallel analyses for all three Anchor models as well as for the Template model. Performing a Monte Carlo-like analysis for both the first Anchor model and the Template model would allow for a more direct comparison with the other two Anchor models. These analyses could also be used to verify whether the differential sensitivity analysis presented in Supplementary Section S5 is consistent with finite changes.

Parameter Randomization Methodology: A subsection should be added to the Methods section detailing how the randomization of parameter values is conducted. This would improve transparency and replicability.

Annotated Equations and Figures: Please address the annotated corrections for the equations and figures/figure captions provided in the attached PDF.

Reviewer #2: The authors have satisfactorily addressed all of my concerns. Thank you for your efforts. The revised manuscript is a significant improvement over the original version, although it is now somewhat lengthy.

I have only a few minor comments:

1. Does the T&A modeling paradigm require a time-scale separation between the dynamics at the template level and those at the anchor level? The example provided seems to use this separation, where the dynamics or response at the anchor level is faster compared to the template level, allowing it to be approximated as being at a steady state.

2. When calibrating dose responses to data, it would be helpful to include the data on the associated plots. This would allow readers to assess the quality of the fitting more effectively.

**Have the authors made all data and (if applicable) computational code underlying the findings in their manuscript fully available?**

Reviewer #1: Yes

Reviewer #2: Yes

PLOS authors have the option to publish the peer review history of their article (what does this mean? ). If published, this will include your full peer review and any attached files.

**Do you want your identity to be public for this peer review?** For information about this choice, including consent withdrawal, please see our Privacy Policy .

Reviewer #1: No

Reviewer #2: No

---

## [Editor Report · Decision Letter 3]

31 Jan 2025

Dear Dr. Voit,

We are pleased to inform you that your manuscript 'Analysis of Systemic Effects of Dioxin on Human Health through Template-and-Anchor Modeling' has been provisionally accepted for publication in PLOS Computational Biology.

Best regards,

Pedro Mendes, PhD

Section Editor

PLOS Computational Biology

Pedro Mendes

Section Editor

PLOS Computational Biology

---

## [Editor Report · Acceptance letter]

PCOMPBIOL-D-24-00128R3

Analysis of Systemic Effects of Dioxin on Human Health through Template-and-Anchor Modeling 

Dear Dr Voit,

I am pleased to inform you that your manuscript has been formally accepted for publication in PLOS Computational Biology. Your manuscript is now with our production department and you will be notified of the publication date in due course.

With kind regards,

Livia Horvath
